# The landscape of m$^1$A modification and its posttranscriptional regulatory functions in primary neurons

Chi Zhang[1†], Xianfu Yi[2*†], Mengfan Hou[3†], Qingyang Li[1], Xueying Li[1], Lu Lu[3], Enlin Qi[1], Mingxin Wu[3], Lin Qi[4], Huan Jian[3], Zhangyang Qi[1], Yigang Lv[3], Xiaohong Kong[1], Mingjun Bi[1*], Shiqing Feng[1,3*], Hengxing Zhou[1,3*]

[1]Department of Orthopaedics, Qilu Hospital of Shandong University, Shandong University Centre for Orthopaedics, Advanced Medical Research Institute, Cheeloo College of Medicine, Shandong University, Jinan, China; [2]Department of Bioinformatics, School of Basic Medical Sciences, Tianjin Medical University, Tianjin, China; [3]Department of Orthopaedics, Tianjin Medical University General Hospital, International Science and Technology Cooperation Base of Spinal Cord Injury, Tianjin Key Laboratory of Spine and Spinal Cord, Tianjin, China; [4]Department of Orthopedics, Hunan Key Laboratory of Tumor Models and Individualized Medicine, The Second Xiangya Hospital, Central South University, Changsha, China

*For correspondence:
yixianfu@tmu.edu.cn (XY);
mingjunbi@sdu.edu.cn (MB);
shiqingfeng@sdu.edu.cn (SF);
zhouhengxing@sdu.edu.cn (HZ)

†These authors contributed equally to this work

Competing interest: The authors declare that no competing interests exist.

**Abstract** Cerebral ischaemia–reperfusion injury (IRI), during which neurons undergo oxygen-glucose deprivation/reoxygenation (OGD/R), is a notable pathological process in many neurological diseases. N1-methyladenosine (m$^1$A) is an RNA modification that can affect gene expression and RNA stability. The m$^1$A landscape and potential functions of m$^1$A modification in neurons remain poorly understood. We explored RNA (mRNA, lncRNA, and circRNA) m$^1$A modification in normal and OGD/R-treated mouse neurons and the effect of m$^1$A on diverse RNAs. We investigated the m$^1$A landscape in primary neurons, identified m$^1$A-modified RNAs, and found that OGD/R increased the number of m$^1$A RNAs. m$^1$A modification might also affect the regulatory mechanisms of noncoding RNAs, e.g., lncRNA–RNA binding proteins (RBPs) interactions and circRNA translation. We showed that m$^1$A modification mediates the circRNA/lncRNA–miRNA–mRNA competing endogenous RNA (ceRNA) mechanism and that 3' untranslated region (3'UTR) modification of mRNAs can hinder miRNA–mRNA binding. Three modification patterns were identified, and genes with different patterns had intrinsic mechanisms with potential m$^1$A-regulatory specificity. Systematic analysis of the m$^1$A landscape in normal and OGD/R neurons lays a critical foundation for understanding RNA modification and provides new perspectives and a theoretical basis for treating and developing drugs for OGD/R pathology-related diseases.

## Editor's evaluation

This study presents a rather valuable finding on the exploration of RNAs m1A modification in normal and OGD/R-treated neurons and the effects of m1A on diverse RNAs. The evidence supporting the claims of the authors is quite solid. The work will be of interest to scientists working in the field of m1A modifications. Noteworthy, this manuscript provides a new avenue for the development of novel therapeutics against ODG/R-related disease.

## Introduction

RNA modifications were first identified more than 50 years ago (*Dunn, 1961*). With the development of sequencing technologies, our understanding of the features (location, function, and regulation) of RNA modifications has greatly improved. RNA modifications include N6-methyladenosine (m6A), N1-methyladenosine (m1A), 5-hydroxymethylcytosine (hm5C), 5-methylcytosine (m5C), pseudouridine (Ψ), and inosine (*Yoon et al., 2017*). Several studies have indicated that in addition to m6A, m1A is an abundant epitranscriptomic modification and regulates multiple biological processes ranging from local structural stability (*Oerum et al., 2017*) to RNA–protein interactions (*Zhao et al., 2019*), apoptosis, and cell proliferation (*Li et al., 2016*; *Chen et al., 2019c*).

m1A was first differentiated from other RNA modifications by Dunn in 1961 (*Dunn, 1961*). m1A modification constitutes the addition of a methyl group to the N1 position of adenosine via m1A regulators. Similar to m6A RNA modification, three kinds of regulators mediate m1A status: 'erasers' (ALKBH1 ALKBH3 and FTO) (*Liu et al., 2016*; *Safra et al., 2017*), 'writers' (TRMT10C, TRMT61B, and TRMT6/61 A) (*Chujo and Suzuki, 2012*; *Engel and Chen, 2018*), and 'readers' (YTHDF1, YTHDF2, YTHDF3, and YTHDC1) (*Dai et al., 2018*; *Seo and Kleiner, 2020*). Writers are methyltransferases that manipulate the level of m1A to interfere with translation. Human mitochondrial tRNAs (mt-tRNAs) contain m1A at positions 9 and 58 (*Guelorget et al., 2010*). TRMT61B and TRMT6/61 A catalyse m1A modification at position 58 in mt-RNA in human cells, while TRMT10C catalyses it at position 9 (*Ozanick et al., 2005*; *Chujo and Suzuki, 2012*). Readers recognize m1A and mediate the translation and degradation of downstream RNAs. Proteins containing YTH domains directly bind to m1A-modified positions in RNA transcripts (*Dai et al., 2018*). Erasers are demethylases that catalyze the removal of m1A from single-stranded DNA and RNA (*Liu et al., 2016*). Two AlkB family proteins, ALKBH3 and ALKBH1, have been found to remove m1A (*Ueda et al., 2017*). Recent studies have revealed that m1A modification is involved in various biological functions. Wu et al. found that m1A demethylase ALKBH3 regulated glycolysis of cancer cells and further affected tumour growth and cancer progression (*Wu et al., 2022a*). Wu et al. revealed that m1A regulation is significantly associated with the pathogenesis of human Abdominal Aortic Aneurysm (*Wu et al., 2022b*). However, the roles of m1A modification in neurons remain largely unknown.

Cerebral ischaemia–reperfusion injury (IRI) is the main pathological process in many brain diseases, such as traumatic brain injury (TBI) (*Zhao et al., 2018*) and acute ischaemic stroke (*Wiberg et al., 2016*). Ischaemia and blood flow reperfusion cause damage at ischaemic sites. Neuronal oxygen-glucose deprivation/reoxygenation (OGD/R), which results in dysregulation of material and energy metabolism, subsequently affects neuronal biological processes such as survival (*Sasaki et al., 2011*), apoptosis (*Song et al., 2019*), and autophagy (*Wang et al., 2014*). Dysfunction of biological processes resulting from OGD/R leads to the occurrence and aggravation of diseases (*Wang et al., 2014*; *Wiberg et al., 2016*; *Song et al., 2019*). Therefore, it is highly important to study the changes in neurons undergoing OGD/R. Recently, many studies have shown that various mRNAs and noncoding RNAs (long noncoding RNAs [lncRNAs], circular RNAs [circRNAs], and microRNAs [miRNAs]) regulate the exacerbation or amelioration of pathological processes caused by OGD/R (*Wang et al., 2017*; *Chen et al., 2020*; *Chen et al., 2021*; *Yang et al., 2022*) through various mechanisms. For example, the lncRNA U90926 directly binds to malate dehydrogenase 2 (MDH2), which could aggravate ischaemic brain injury by facilitating neutrophil infiltration (*Chen et al., 2021*). circUCK2 functions as a sponge to inhibit miR-125b-5p activity, resulting in an increase in growth differentiation factor 11 (GDF11) expression and subsequent amelioration of neuronal injury (*Yang et al., 2022*). Some studies have shown that chemical modifications of these RNAs can regulate the original mechanism (*Patil et al., 2016*; *Yang et al., 2018*; *Chen et al., 2019a*; *Wen et al., 2020*; *Xu et al., 2020*; *Liu et al., 2021b*). However, most studies have focused on m6A modification, and little is known about the features and functions of m1A modification of various RNAs in normal neurons and OGD/R-treated neurons.

In this study, we identified the characteristics of m1A modification of various RNAs (mRNA, lncRNA, and circRNA) in mouse neurons and explored the potential effects of this modification on different RNA functions. We first identified m1A-modified peaks (m1A peaks) in normal neurons and OGD/R--treated neurons at different times and thus discovered a GA-rich motif. The number of RNAs with m1A modification, the chromosome distribution, and the changes in m1A modification on various RNAs before and after OGD/R treatment were analysed. The number of m1A RNAs was increased after OGD/R treatment, and most of these differentially expressed m1A-modified RNAs were involved in

biological processes and signalling pathways related to the regulation of cellular homeostasis and synapses. In addition, the results indicated that m1A modification mediates the complex posttranscriptional regulation network and that the regulation of m1A modification in key nodes of RNA interaction networks may cause widespread changes in downstream signalling. We also proposed three patterns of m1A modification, with genes fitting these modification patterns participating in different biological functions and signalling pathways. Overall, we analysed the characteristics of m1A modification of multiple RNAs in normal neurons and OGD/R-treated neurons from multiple perspectives. The landscape and other findings provide evidence useful for the exploration of epitranscriptomic mechanisms and the development of targeted drugs in OGD/R-related pathological processes.

## Results

### The common features of m1A modification in primary neurons

The distribution of m1A modifications in the transcriptome is uncharacterized in the nervous system, especially in important component neurons. We performed m1A RNA immunoprecipitation sequencing (MeRIP-seq) to clarify the m1A transcriptomic landscape in primary neurons and OGD/R-treated neurons (*Figure 1A*). A total of 48,260, 13,588, and 16,397 m1A peaks were identified in mRNAs, lncRNAs, and circRNAs, respectively (*Figure 1B*). For example, regarding the m1A peaks in mRNAs, 29,066, 37,893, and 41,334 m1A peaks were identified in the Control, OGD/R 1.5 hr, and OGD/R 3 hr samples, respectively (*Figure 1—figure supplement 1A*). We examined the genomic distribution of those m1A peaks and found that in the Control group, 54.2% were located in coding sequences (CDSs) and 25.1% were located in the 5' untranslated region (5'UTRs) in mRNAs, with 88.4% located in CDSs and 8.9% in 5'UTRs in circRNAs (*Figure 1C*). A similar proportion was found in the OGD/R 1.5 hr and 3 hr groups (*Figure 1—figure supplement 1B*). The genomic distribution pattern was slightly different from previous findings (*Dominissini et al., 2016*). However, more m1A peaks were in the CDS in circRNAs than in mRNAs (88.4% vs 54.2% in the Control group), and a clear decreasing trend in near-5'UTRs was observed (*Figure 1D* and *Figure 1—figure supplement 1C*). The m1A genomic distribution in mRNAs and circRNAs differed significantly between normal and OGD/R-treated neurons (chi-squared test, p<0.05, *Figure 1—figure supplement 1B*). The above results indicated the presence of abundant m1A modifications in neurons and showed that different OGD/R treatments may affect these modifications.

Some studies have shown that m1A motifs, such as GUUCNANNC motifs (*Safra et al., 2017*), GUUCRA motifs (*Han et al., 2017*), and GA-rich consensus sequences (*Li et al., 2016*), exist in different cells. However, there are no accepted highly conserved regions for m1A modification. We used unbiased motif detection to reveal the potential motifs for m1A peaks with MEME Suite (*Bailey et al., 2015*) (v5.4.1). Various motifs that were also GA-rich were identified in mRNAs and circRNAs (*Figure 1E* and *Figure 1—figure supplement 1D*). Indeed, more than 90% (4510/4646 in the Control group) of motifs in mRNA were GA-rich, and approximately 10% (579/4391 in the Control group) of motifs in circRNA were GA-rich. This pattern indicated that GA-rich m1A motifs are more prevalent in mRNA and that various mechanisms may control m1A modification in circRNA. By comparing these motifs with the known motifs in the JASPAR database (*Castro-Mondragon et al., 2022*) (v 2022), we found that the GA-rich motifs are highly similar to those in MA0528.1 from ZNF263 (*Figure 1E* and *Figure 1—figure supplement 1D*), and the functions of these motifs were mainly focused on RNA polymerase II promoter regulation. Interestingly, although the motifs in circRNAs in the OGD/R 3 hr group were also GA-rich, their main function was to regulate axon guidance and the G-protein coupled receptor protein signalling pathway. These results suggest that GA-rich m1A motifs are present in neurons, although they vary across RNAs with different functions.

The dynamic regulation and function of m1A mainly depend on m1A demethylase ('erasers'), m1A methyltransferase ('writers'), and m1A read-binding protein ('readers'). We wondered whether these m1A regulators were differentially expressed after OGD/R-treatment. The results showed that the expression of *Alkbh3*, *Trmt10c*, *Trmt61a*, *Ythdf2*, and *Ythdf3* was statistically different among the three groups (*Figure 1F*). However, the trends of these m1A regulators were different under different OGD/R treatments. This result implies that the m1A modification process in neurons may be a very complex process, and the specific mechanisms involved in this process still need to be further explored.

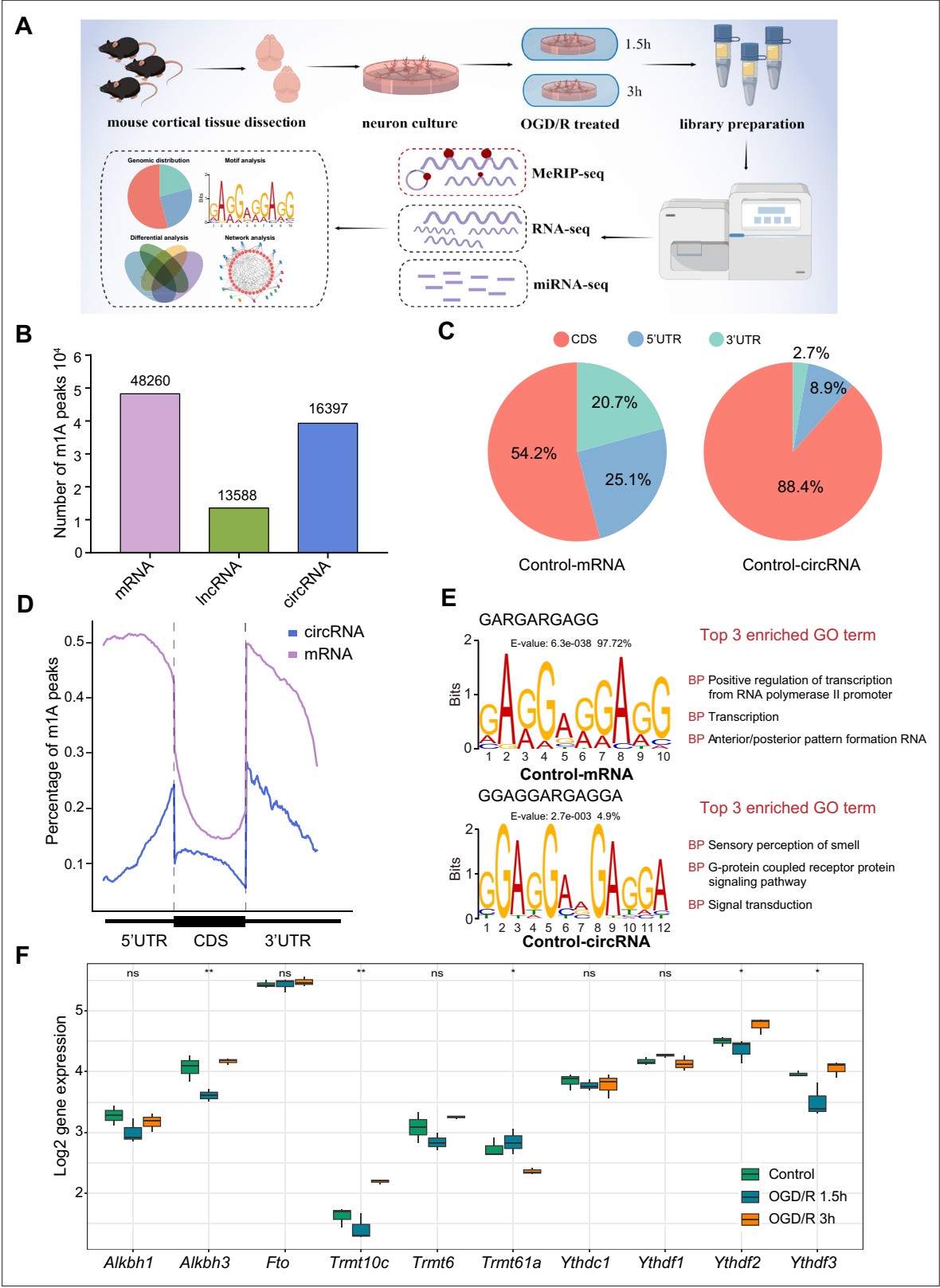

**Figure 1.** The common m¹A modification features in primary neurons. (**A**) Schematic of the experimental design and data analysis workflow (drawn by Figdraw). (B) m¹A peaks identified in different RNAs (mRNA, lncRNA, and circRNA). (C) The genomic distribution of m¹A peaks in mRNAs and circRNAs. (D) The genomic distribution pattern of m¹A peaks in mRNAs and circRNAs on a metagene. (E) Potential motifs for m¹A peaks in both m¹A mRNAs and m¹A circRNAs in the Control group. The E-value of a motif is based on its log likelihood ratio, width, sites, and background letter frequencies. The

*Figure 1 continued on next page*

*Figure 1 continued*

percentage represents the proportion of a motif in all identified motifs. (F) Boxplots showing differential gene expression of m$^1$A regulators among three groups, One-way ANOVA was used to compare the means among different groups (* p<0.05, ** p<0.01, *** p<0.001, n=3 biologically independent samples per group).

The online version of this article includes the following figure supplement(s) for figure 1:

**Figure supplement 1.** The common m$^1$A modification features in primary neurons.

## OGD/R increases the number of m$^1$A mRNAs and affects neuron fate

m$^6$A modifications can functionally alter the expression of mRNAs, pre-mRNAs, miRNAs, and noncoding RNAs, such as rRNAs and tRNAs (*Chua et al., 2020*). Although m$^1$A is another abundant RNA modification, the characteristics of this modification on mRNAs and noncoding RNAs in neurons remain unclear. We identified m$^1$A modifications on different kinds of RNAs (mRNAs, lncRNAs, and circRNAs) in normal primary neurons and OGD/R-treated neurons to determine whether there are characteristic modifications of different RNAs after OGD/R treatment. First, we determined the numbers of m$^1$A mRNAs identified by four different methods: the conventional, treatment, mismatch, and trough methods (see the Materials and methods section for details). The most m$^1$A mRNAs were identified by the conventional method, while the mismatch method identified the least (*Figure 2—figure supplement 1A*). We also identified the unique and common m$^1$A mRNAs in different OGD/R-treated neurons by four different methods (*Figure 2—figure supplement 1B*). The percentages of common (5.82–76.44%) and unique mRNAs (8.48–70.1% in the Control group, 11.56–74.17% in the OGD/R 1.5 hr group, and 11.59–76.03% in the OGD/R 3 hr group) identified by the different methods varied greatly. To ensure the accuracy of subsequent analyses as much as possible, we selected the m$^1$A mRNAs identified by at least two of the abovementioned methods for subsequent analyses (*Figure 2—figure supplement 1C*), resulting in 4047, 5756, and 4948 (*Figure 2—source data 1*) m$^1$A mRNAs for the Control, OGD/R 1.5 hr, and OGD/R 3 hr groups, respectively (*Figure 2A*).

Then, we used density distribution (*Figure 2B*, left) and cumulative distribution function curves (*Figure 2B*, right) to compare the m$^1$A level among the groups. After OGD/R treatment, the m$^1$A level in neurons was higher than that in the Control group (OGD/R 1.5 hr vs Control: p<1.1e-15, OGD/R 3 hr vs Control: p<2.2e-16, Kolmogorov–Smirnov test). Since the modification density differed by treatment, we sought to determine whether the amount of m$^1$A modification on each chromosome differs before and after treatment. The numbers of m$^1$A mRNAs on each chromosome were increased after OGD/R treatment (*Figure 2C*).

Finally, differential methylation analysis was conducted between OGD/R-treated neurons and normal neurons (*Figure 2—source data 2*). During the different OGD/R treatments, the mRNAs with large changes in methylation levels were not totally consistent between the two groups, possibly implying that distinct m$^1$A modification patterns exist in the different OGD/R-treated groups (*Figure 2D*). To further clarify the functions of the differentially m$^1$A methylated mRNAs, GO and KEGG enrichment analyses were performed. The m$^1$A mRNAs with decreased methylation in the OGD/R 1.5 hr group regulated synapses, the release of neurotransmitters, the cAMP signalling pathway, and axon guidance (*Figure 2E*). In contrast, the m$^1$A mRNAs with decreased methylation in the OGD/R 3 hr group regulated synaptic plasticity, the membrane potential, glutamatergic synapses, and the calcium signalling pathway (*Figure 2F*). The m$^1$A mRNAs with increased methylation in the OGD/R 1.5 hr group influenced morphogenesis, cell fate commitment, chemical carcinogenesis, and apoptosis, while the m$^1$A mRNAs with increased methylation in the OGD/R 3 hr group affected transmembrane receptors, cell substrate adhesion, focal adhesion, and regulation of the actin cytoskeleton (*Figure 2—figure supplement 1D–E*). The above results indicate that under different OGD/R treatments, the increases and decreases in m$^1$A in neurons affect different biological functions. With prolonged OGD/R time, differential m$^1$A modification regulates the fate of neurons.

## OGD/R increases the number of m$^1$A lncRNAs and affects RNA processing

m$^6$A modification has been reported to be present on lncRNAs and to affect their biological functions (*Patil et al., 2016*; *Yang et al., 2018*; *Wen et al., 2020*). However, the m$^1$A modification of lncRNAs and its possible role in neurons remain uncharacterized. We identified lncRNAs with m$^1$A modification in

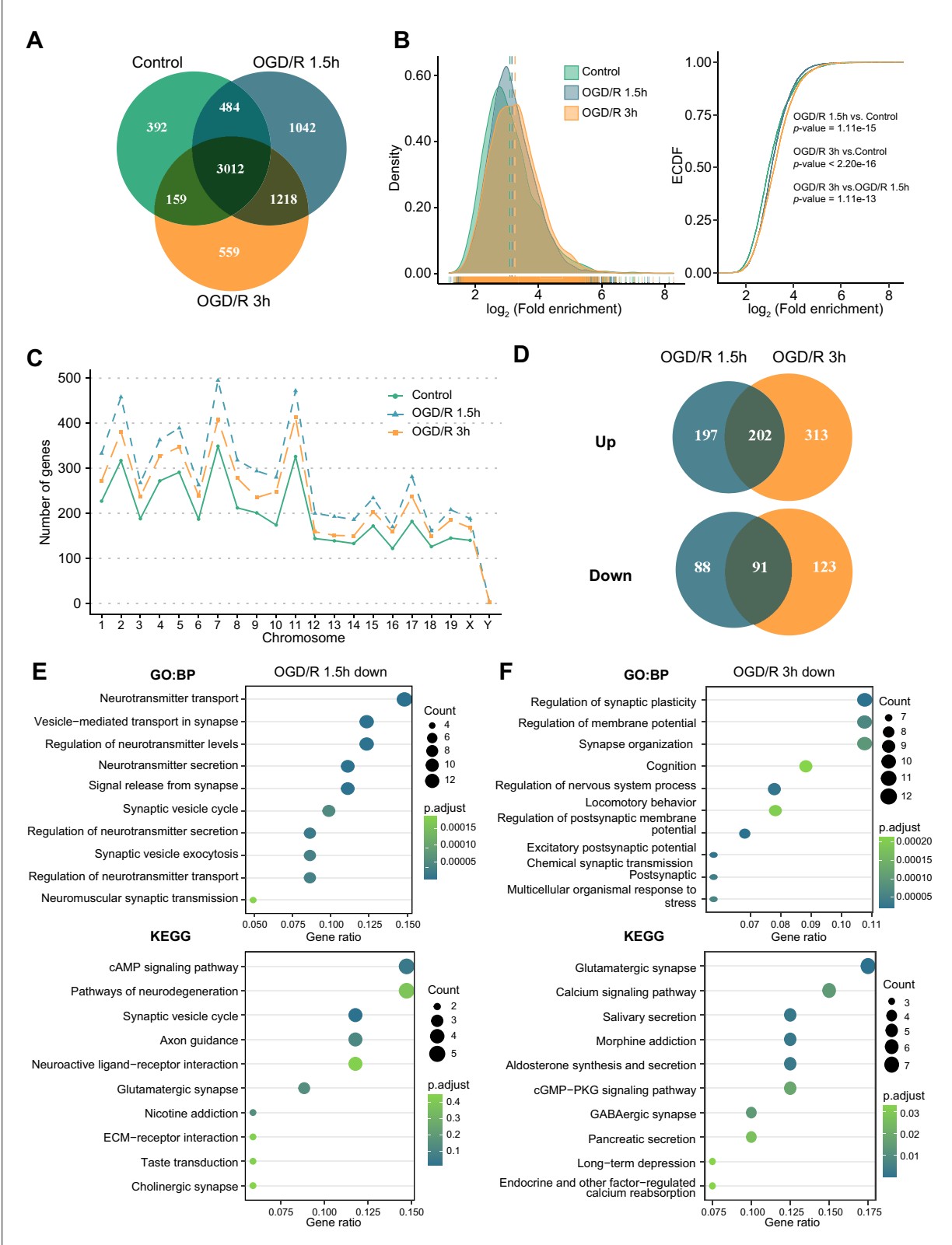

**Figure 2.** OGD/R increases the number of m¹A mRNAs and affects neuron fate. (**A**) Shared and unique m¹A mRNAs in the Control and different OGD/R groups. B. The density distribution (left) and cumulative distribution function curves (right) to show the m¹A modification level among different groups. The Kolmogorov−Smirnov test was used to test the significance. C. The number of m¹A mRNAs on each chromosome in different groups. D. Venn

*Figure 2 continued on next page*

*Figure 2 continued*

diagram showing differentially methylated (up and down) m$^1$A mRNAs in the OGD/R 1.5 hr and OGD/R 3 hr groups. E-F. GO and KEGG analyses of differentially methylated (down) m$^1$A mRNAs in the OGD/R 1.5 hr (**E**) and OGD/R 3 hr (**F**) groups.

The online version of this article includes the following source data and figure supplement(s) for figure 2:

**Source data 1.** Modification sites detected for mRNAs in three groups.

**Source data 2.** Differential modification sites detected for mRNAs between OGD/R and Control.

**Figure supplement 1.** OGD/R increases the number of m$^1$A mRNAs and affects neuron fate.

---

normal neurons and OGD/R-treated neurons to explore the potential functions of m$^1$A lncRNAs. First, we applied the four abovementioned methods to identify m$^1$A lncRNAs in normal and OGD/R-treated neurons. The conventional method identified the most m$^1$A lncRNAs (*Figure 3—figure supplement 1A*). We then identified the unique and common m$^1$A lncRNAs in different OGD/R-treated neurons by the four different methods (*Figure 3—figure supplement 1B*). Fewer m$^1$A lncRNAs than m$^1$A mRNAs were identified by the four methods (*Figure 3—figure supplement 1C*). The m$^1$A lncRNAs identified by at least two methods were selected for further analysis (Control: 1078, OGD/R 1.5 hr: 1661, OGD/R 3 hr: 1294) (*Figure 3A*, *Figure 3—figure supplement 1D* and *Figure 3—source data 1*).

We next analysed the genomic sources of m$^1$A lncRNAs in the Control and OGD/R-treated groups. The two main genomic sources of m$^1$A lncRNAs were exon sense-overlapping regions and intergenic regions (*Figure 3B*). This distribution did not differ significantly between the Control and OGD/R groups (chi-squared test, p=0.303). Similar to the mRNA modification characteristics (*Figure 2B*), the differences in the lncRNA m$^1$A modification level between the Control and OGD/R-treated groups were statistically significant (*Figure 3C*, OGD/R 1.5 hr vs Control: p<0.001, OGD/R 3 hr vs Control: P<2.508e-5; Kolmogorov–Smirnov test). We also counted the m$^1$A lncRNAs on each chromosome in normal and OGD/R-treated neurons and found that the number of m$^1$A lncRNAs on each chromosome was increased after OGD/R treatment (*Figure 3D*).

Interaction with RBPs is an important regulatory mechanism of lncRNAs (*Yang et al., 2020a*; *Søndergaard et al., 2022*). We thus sought to determine whether lncRNAs with m$^1$A modifications play vital roles in biological processes, such as RNA processing. Considering the differentially m$^1$A-methylated lncRNAs in the OGD/R groups (*Figure 3—figure supplement 1E*) (*Figure 3—source data 2*), RBPs binding to these lncRNAs with high reliability were predicted in the ENCORI database, and lncRNA–RBP interaction networks were then constructed (*Figure 3E*). In the OGD/R 1.5 hr and OGD/R 3 hr lncRNA–RBP networks, *Meg3* and *Neat1*, which have been reported to play an important role in nervous system-related diseases, were predicted to bind to various RBPs (*Zhong et al., 2017*; *Sanli et al., 2018*; *Cui et al., 2019*; *Liang et al., 2020*). Functional enrichment analysis showed that these RBPs participated in the processes of mRNA metabolism, mRNA processing, and mRNA splicing (*Figure 3F*). These results suggested that the ability of nerve-related lncRNAs to regulate RNA metabolism and then affect the pathophysiological processes in neurological diseases might depend on their m$^1$A modification.

## OGD/R increases the number of m$^1$A modification sites in circRNAs and regulates translation functions

As RNA molecules regulate diverse pathophysiological processes, circRNAs are specifically enriched in the nervous system (*Gokool et al., 2020*; *Mehta et al., 2020*). Some studies have demonstrated that m$^6$A modification can affect circRNA biogenesis, immunogenicity, translation, etc. (*Chen et al., 2019b*; *Tang et al., 2020*; *Xu et al., 2020*). However, the features of m$^1$A modification of circRNAs in the nervous system remain unknown. We investigated m$^1$A modifications in neuronal circRNAs using various approaches. The four abovementioned methods identified different numbers of m$^1$A circRNAs in each group, with the most m$^1$A circRNAs identified by the conventional method (*Figure 4—figure supplement 1A*). We then examined the common and unique m$^1$A circRNAs in the normal and OGD/R-treated groups identified by the four methods. The fewest associated m$^1$A circRNAs were identified by the mismatch method (*Figure 4—figure supplement 1A–B*). Fewer m$^1$A circRNAs than m1A mRNAs were identified by all four methods (*Figure 4—figure supplement 1C*). However, m$^1$A circRNAs were more abundant than m$^1$A lncRNAs. To further investigate the role of m$^1$A circRNAs in

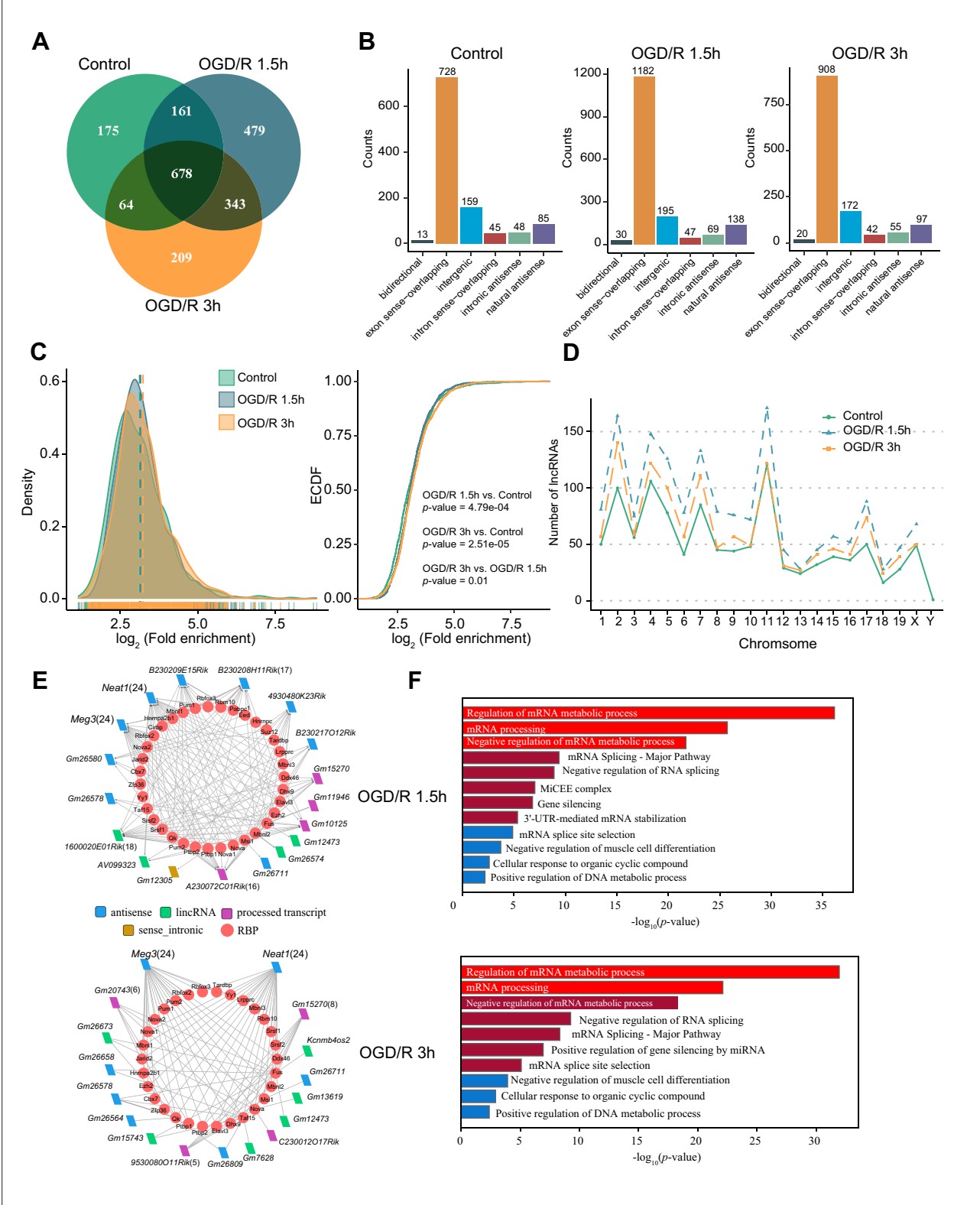

**Figure 3.** OGD/R increases the number of m[1]A lncRNAs and affects RNA processing. (**A**) Shared and unique m[1]A lncRNAs in the Control and different OGD/R groups. (**B**) The genomic resources of m[1]A lncRNAs in the Control and OGD/R-treated groups. (**C**) The density distribution (left) and cumulative distribution function curves (right) show the m[1]A modification level among the different groups. The Kolmogorov–Smirnov test was used to test the significance. (**D**) The number of m[1]A lncRNAs on each chromosome in different groups. (**E-F**) The interaction networks between lncRNAs and RBPs in different OGD/R groups (**E**) and functional enrichment analysis of those core RBPs (**F**).

*Figure 3 continued on next page*

Figure 3 continued

The online version of this article includes the following source data and figure supplement(s) for figure 3:

**Source data 1.** Modification sites detected for lncRNAs in three groups.

**Source data 2.** Differential modification sites detected for lncRNAs between OGD/R and Control.

**Figure supplement 1.** OGD/R increases the number of m$^1$A lncRNAs and affects RNA processing.

each group, we selected the m$^1$A circRNAs identified by at least two methods for downstream analysis (*Figure 4A*, *Figure 4—figure supplement 1D* and *Figure 4—source data 1*).

The genomic sources of those m$^1$A circRNAs were counted in each group, and exonic regions and sense-overlapping regions were found to be the main sources (chi-squared test, p=0.9379) (*Figure 4B*). Next, we generated density distribution and cumulative distribution function curves to explore the m$^1$A modification degree in the three groups. The m$^1$A modification level was increased after OGD/R treatment, and the difference between the OGD/R 3 hr and Control groups was statistically significant (OGD/R 3 hr vs Control p<0.0009, Kolmogorov–Smirnov test). However, the m$^1$A level did not differ significantly between the OGD/R 1.5 hr and Control groups (*Figure 4C*). m$^1$A circRNAs on each chromosome were also analysed, and the numbers of m$^1$A circRNAs were increased after the different OGD/R treatments (*Figure 4D*).

To clarify the biological functions of the differentially m$^1$A-methylated circRNAs in the OGD/R 1.5 hr and OGD/R 3 hr groups (*Figure 4—source data 2*), we obtained the source genes of these differentially methylated circRNAs and performed GO and KEGG functional enrichment analyses. In the OGD/R 1.5 hr group, the differentially methylated circRNAs with decreased m$^1$A levels were enriched in synapse organization, regulation of membrane potential, the cAMP signalling pathway, and the neurodegeneration pathway, while the circRNAs with increased m$^1$A levels were enriched in morphogenesis, GTPase activity, axon guidance, and the MAPK signalling pathway (*Figure 4E* and *Figure 4—figure supplement 1E*). In the OGD/R 3 hr group, the differentially methylated circRNAs with decreased m$^1$A levels were enriched in synaptic plasticity, neurotransmitter transport, glutamatergic synapse, and dopaminergic synapse, while the circRNAs with increased m$^1$A levels were enriched in supramolecular fibre organization, cellular component disassembly, adherens junction, and endocytosis (*Figure 4F* and *Figure 4—figure supplement 1F*). These results indicate that the differentially modified circRNAs regulate different biological functions in different OGD/R processes. The m$^1$A circRNAs with decreased m$^1$A levels were related mainly to synapses and the release of neurotransmitters, while the m$^1$A circRNAs with increased m$^1$A levels played regulatory roles in many biological processes.

Currently, research on the translation ability and products of circRNAs is gradually increasing. We asked whether the m$^1$A circRNAs could be translated. By using the riboCIRC database (*Li et al., 2021*) (*http://www.ribocirc.com/index.html*), we identified some circRNAs (237/811) with translation ability among the identified differentially m$^1$A-modified circRNAs. In addition, m$^6$A modifications annotated in the database were present on some m$^1$A circRNAs. The polypeptide structures were predicted to show the possible translation products. Then, mmu_circ_0000705 (encoded by *App*) and mmu_circ_0002207 (encoded by *Foxo3*) were selected to show the abovementioned features (*Figure 4G–H*, *Figure 4—figure supplement 1G–H*). *App* can regulate the stability of synapses that bind to diverse proteins and regulates the occurrence and development of nervous system diseases (*Lee et al., 2020*; *Eysert et al., 2021*). mmu_circ_0000705 is composed of 5 exons. In this circRNA, m$^6$A modification occurs in the 85025441–85025456 region, while m$^1$A modification occurs in the 85043575–85043600 region. *Foxo3* also regulates multiple functions of the nervous system (*Deng et al., 2018*; *Du et al., 2021*). mmu_circ_0002207 is composed of 1 exon, with m$^6$A modification occurring in the 42196615–42196629 region and m$^1$A modification occurring in the 42196961–42197760 region. These results suggest that m$^1$A modification may play a regulatory role in the translation of circRNAs into peptides. However, the specific mechanism needs further exploration.

## m$^1$A modification affects the ceRNA mechanism of differentially methylated lncRNAs and circRNAs

The ceRNA mechanism is a main posttranscriptional regulatory mechanism of circRNAs and lncRNAs. Studies have shown that the ceRNA mechanism plays an important role in nervous system diseases

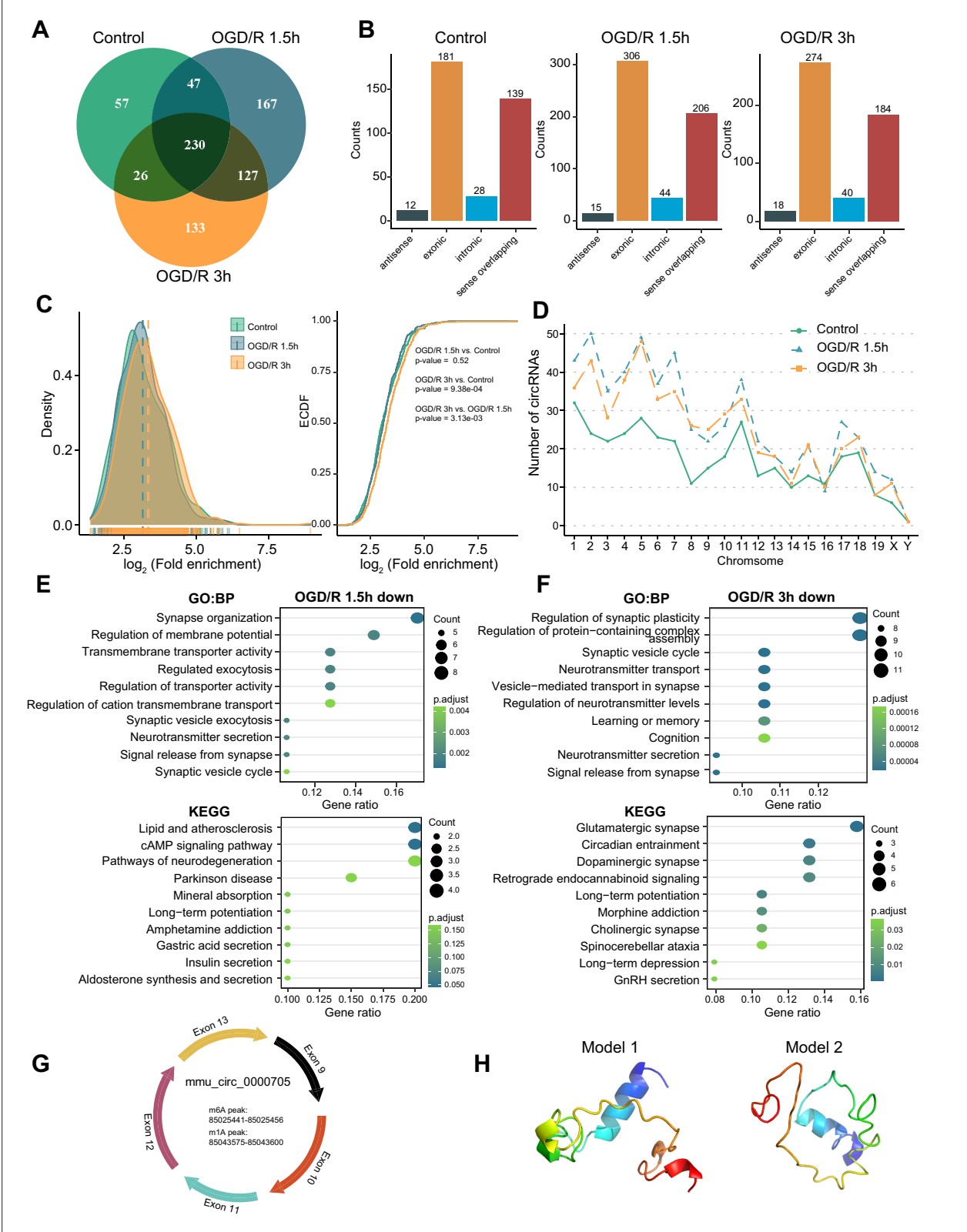

**Figure 4.** OGD/R increases the m[1]A modification sites on circRNA and regulates translation functions. (**A**) Shared and unique m[1]A circRNAs in the Control and different OGD/R groups. (**B**) The genomic resources of m[1]A circRNAs in the Control and OGD/R-treated groups. (**C**) The density distribution (left) and cumulative distribution function curves (right) show the m[1]A modification level among the different groups. The Kolmogorov–Smirnov test was used to test the significance. (**D**) The number of m[1]A circRNAs on each chromosome in different groups. (E-F) GO and KEGG analysis of differentially

*Figure 4 continued on next page*

Figure 4 continued

methylated (down) m1A circRNAs in OGD/R 1.5 hr (**E**) and OGD/R 3 hr (**F**). (G) An example of a circRNA (mmu_circ_0000705) with translation ability that also contains an m$^1$A and m$^6$A modification site. (H) Predicted polypeptide structure of mmu_circ_0000705.

The online version of this article includes the following source data and figure supplement(s) for figure 4:

**Source data 1.** Modification sites detected for circRNAs in three groups.

**Source data 2.** Differential modification sites detected for circRNAs between OGD/R and Control.

**Figure supplement 1.** OGD/R increases the m$^1$A modification sites on circRNA and regulates translation functions.

---

(*Huang et al., 2020*; *Moreno-García et al., 2020*). However, the function and physiological effect of the m$^1$A modifications in these noncoding RNAs remain unclear. We identified the differentially expressed miRNAs in the OGD/R-treated groups; 69 differentially modified miRNAs were found in the OGD/R 1.5 hr group, and 81 differentially modified miRNAs were found in the OGD/R 3 hr group (*Figure 5A* and *Figure 5—figure supplement 1A*). Differentially expressed mRNAs in the OGD/R--treated groups were also identified by RNA-seq. A total of 1579 and 3259 differentially expressed mRNAs were identified in the OGD/R 1.5 hr and OGD/R 3 hr groups, respectively (*Figure 5B* and *Figure 5—figure supplement 1B*). GO and KEGG enrichment analyses were conducted to explore the biological functions of the differentially expressed mRNAs (*Figure 5C*). In the OGD/R 1.5 hr group, the differentially expressed mRNAs were enriched in extracellular matrix organization, the PI3K−Akt signalling pathway, neuroactive ligand−receptor interaction, and the calcium and Hippo signalling pathways. In the OGD/R 3 hr group (*Figure 5—figure supplement 1C*), the differentially expressed mRNAs were enriched in synapse organization, axonogenesis, regulation of membrane potential, regulation of neurogenesis, etc. The functions of the differentially expressed genes were mainly related to the influence on cell structure in the OGD/R 1.5 hr group but to synapse, axon, and other neuron functions in the OGD/R 3 hr group.

Based on the differentially expressed miRNAs (*Figure 5—source data 1*), we performed a screen to identify the differentially expressed and differentially methylated mRNAs, lncRNAs, and circRNAs that bind to these miRNAs. Then, the identified miRNAs, mRNAs, lncRNAs, and circRNAs were used to construct expression- and methylation-specific ceRNA regulatory networks. In the OGD/R 1.5 hr expression ceRNA network, *H19*, a well-known lncRNA, was shown to regulate circRNAs by sponging miRNAs (*Figure 5D*, left and *Figure 5—figure supplement 1D*, top). In the OGD/R 3 hr expression ceRNA network, *Neat1* and *Rian* were shown to regulate various miRNAs and affect downstream mRNA and circRNA expression (*Figure 5D*, right and *Figure 5—figure supplement 1D*, bottom). In the OGD/R 1.5 hr methylation ceRNA network, *Neat1*, which plays critical regulatory roles in diverse neurological diseases, was shown to sponge many miRNAs and affect other lncRNA/circRNA−miRNA axes (*Figure 5E*, left and *Figure 5—figure supplement 1E*). In the OGD/R 3 hr methylation ceRNA network, *Neat1* was also shown to play important roles (*Figure 5E*, right). Interestingly, the miRNAs sponged by *Neat1* in the methylation ceRNA network were different from those in the expression ceRNA network. *Neat1*, present in both the expression and methylation ceRNA networks, sponged different miRNAs in the two networks, suggesting that m$^1$A modification of lncRNAs impacts the ceRNA mechanism.

## m$^1$A modification of mRNA 3′UTRs hinders miRNA−mRNA binding

Our above analysis indicated that m$^1$A modification may affect the ceRNA mechanism of different RNAs. miRNA binding sites in mRNAs are located mainly in the 3'UTR. Therefore, we next explored the changes in miRNA−mRNA pairs under different OGD/R treatments. By analysing the m$^1$A-modified peak regions and the positions of miRNA binding seed sequences, we found that there were some miRNA−mRNA pairs that changed dynamically during different OGD/R treatments (*Figure 5F*). Among these miRNA−mRNA pairs, some mRNAs have been shown to be associated with neural tissue development (*Rbfox3*) (*Kim et al., 2013*), neural progenitor differentiation (*Arid1a*) (*Liu et al., 2021a*), NMDA receptor complexes (*Grin2d*) (*XiangWei et al., 2019*), and ubiquitination of proteins (*Wdtc1*) (*Groh et al., 2016*). Therefore, for these key miRNA−mRNA pairs, we designed a dual lucif-erase assay to detect whether m$^1$A modification of the 3'UTR affects the binding of miRNAs to mRNAs (*Figure 5—source data 2*). *Arid1a* encodes a member of the SWI/SNF family that belongs to the neural progenitor-specific chromatin remodelling complex (npBAF complex) and the neuron-specific

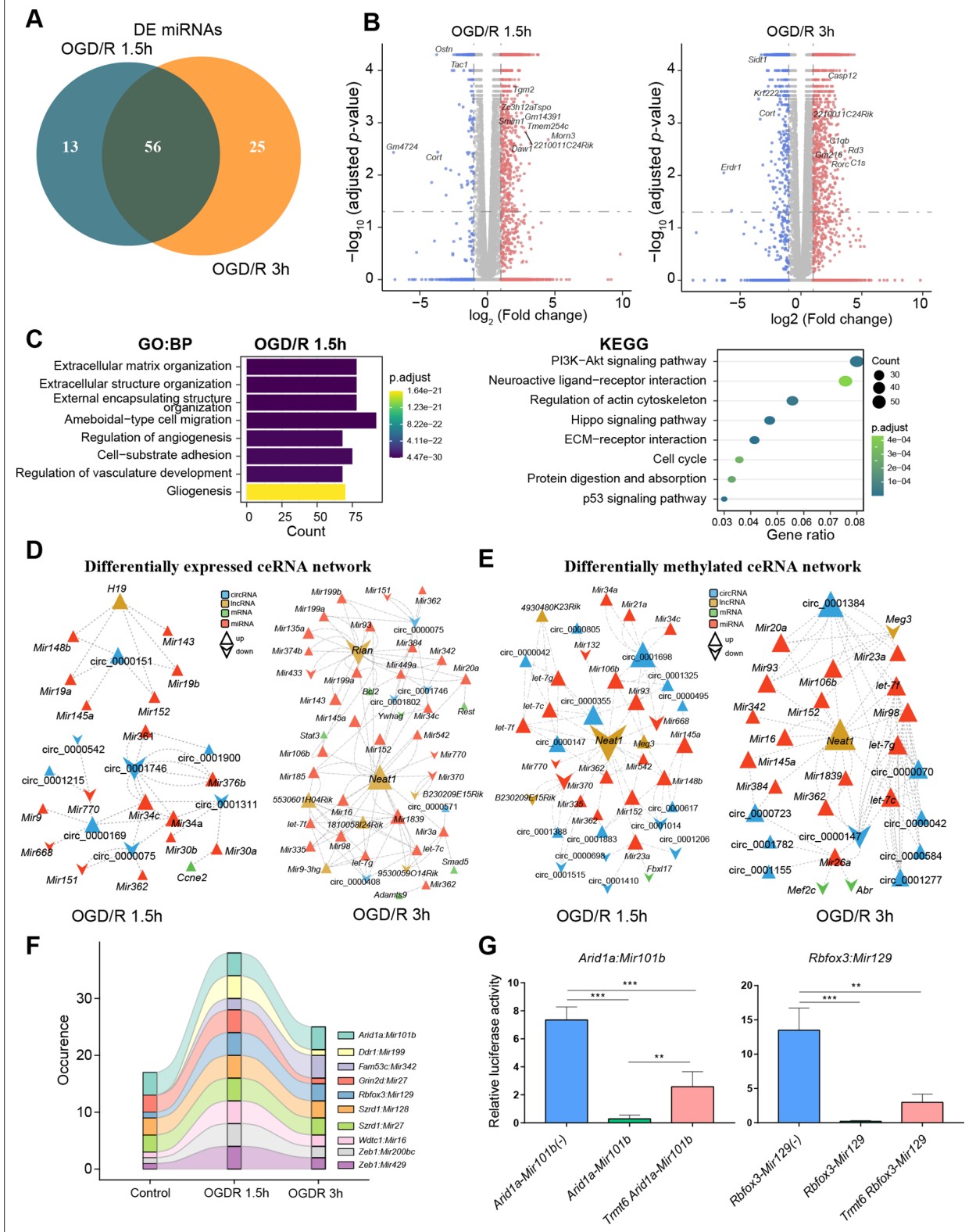

**Figure 5.** m1A modification affects the ceRNA mechanism of differentially methylated lncRNAs and circRNAs. (**A**) Venn diagram showing differentially expressed miRNAs in different OGD/R-treated groups. (B) Volcano plot showing differentially expressed mRNAs in different OGD/R-treated groups (OGD/R 1.5 hr, left and OGD/R 3 hr, right). (C) GO analysis shows the biological functions of the differentially expressed mRNAs in the OGD/R 1.5 hr group. D-E. Expression- (**D**) and methylation-specific (**E**) ceRNA regulatory networks in the OGD/R 1.5 hr and OGD/R 3 hr groups. (F) Sankey diagram showing the dynamic changes in miRNA–mRNA pairs in the Control, OGD/R 1.5 hr and OGD/R 3 hr groups. (G) A dual luciferase assay showed that

*Figure 5 continued on next page*

*Figure 5 continued*

m1A modification of *Arid1a* and *Rbfox3* blocks the binding of the corresponding miRNAs. Sample size = 3 for each group, error bar represents the standard deviation of triplicate measurements. One-way ANOVA for comparison among three groups. * p < 0.05, ** p < 0.01, *** p < 0.001, **** p < 0.0001.

The online version of this article includes the following source data and figure supplement(s) for figure 5:

**Source data 1.** Differential expression of miRNAs between OGD/R and Control.

**Source data 2.** The miRNA sequences and mRNA sequences used in *Figure 5*.

**Figure supplement 1.** m$^1$A modification affects the ceRNA mechanism of differentially methylated lncRNAs and circRNAs.

chromatin remodelling complex (nBAF complex). We cotransfected *Arid1a*, *Mir101b,* and *Trmt6* (an m$^1$A methyltransferase) into HEK293T cells. *Trmt6* alleviated the inhibitory effect of *Mir101b* on *Arid1a* expression (one-way ANOVA, p<0.05). *Rbfox3–Mir129* is another miRNA–mRNA pair affected by *Trmt6*. After cotransfection of *Rbfox3–Mir129* and *Trmt6*, *Trmt6* alleviated the repression of target genes by the miRNA (albeit not significantly, one-way ANOVA) (*Figure 5G*). Regarding the other two genes (*Grin2d* and *Wdtc1*), after cotransfection with their corresponding miRNAs and *Trmt6*, *Trmt6* did not affect the ability of the miRNAs to bind to their target genes (*Figure 5—figure supplement 1F*). This result suggests that m$^1$A modification of the 3'UTR in some mRNAs does affect the binding of miRNAs. The underlying mechanisms appear complex and need further exploration.

## Three patterns of m$^1$A modification regulation in neurons

We identified features of m$^1$A modifications in mRNAs and noncoding RNAs and explored the potential functions of those RNAs. However, the m$^1$A patterns remain unknown. We applied the NMF method to identify m$^1$A modification patterns; $k=3$ was the value at which the largest change occurred and was thus selected as the optimal $k$ value (*Figure 6—figure supplement 1A*). Three m$^1$A modification patterns, termed Cluster 1, Cluster 2, and Cluster 3, were discovered (*Figure 6A*). The mixture coefficient matrix also showed that the three clusters had good discrimination (*Figure 6B*). We further explored whether the different m$^1$A clusters had different biological functions. Cluster 1 was defined as the 'Metabolism-associated cluster' (MAC) since its members were enriched in signal transduction and biosynthetic processes (*Figure 6C* and *Figure 6—figure supplement 1B*) and metabolism-associated pathways, such as the chemokine and relaxin signalling pathways. In Cluster 2, autophagy-related biological processes and pathways (*Figure 6D* and *Figure 6—figure supplement 1C*) were enriched; thus, we defined Cluster 2 as the 'Autophagy-associated cluster' (AAC). Finally, Cluster 3 was defined as the 'Catabolism-associated cluster' (CAC) because of the enrichment of catabolic processes (*Figure 6E* and *Figure 6—figure supplement 1D*). The above results show that m$^1$A modification plays different regulatory roles in the different clusters, indicating the existence of specific m$^1$A modification patterns in neurons. To further explore the upstream regulatory mechanisms of these three m$^1$A patterns, we performed a transcription factor (TF) analysis. The 'Enrichr' database (http://amp.pharm.mssm.edu/Enrichr/) was used to enrich the TFs (*Figure 6—figure supplement 1E–G*). We found that the representative TFs enriched specifically in clusters. Further prediction of the activity of these transcription factorTFs showed that different TF activation and repression patterns exist in different clusters. In MAC and CAC, most of the TFs had increased inferred activity. However, most of the TFs had decreased inferred activity in AAC (*Figure 6F–H*). Based on these TFs with altered activity, we constructed a TF-target gene regulatory network (*Figure 6I–K*). The above results suggested that the three m$^1$A patterns we identified are indeed regulated by different TFs and that the activities of these TFs are also specific. Another question we are interested in is whether these three different functional m$^1$A patterns are associated with the expression of m$^1$A regulators. We calculated the scores of representative functions in the three m$^1$A patterns and then calculated the correlation between these functional scores and the expression of m$^1$A regulators. Our results suggested that most of m$^1$A regulators were positively correlated with metabolism-associated functions and autophagy-associated functions, and negatively correlated with catabolism-associated functions (*Figure 6—figure supplement 1H*).

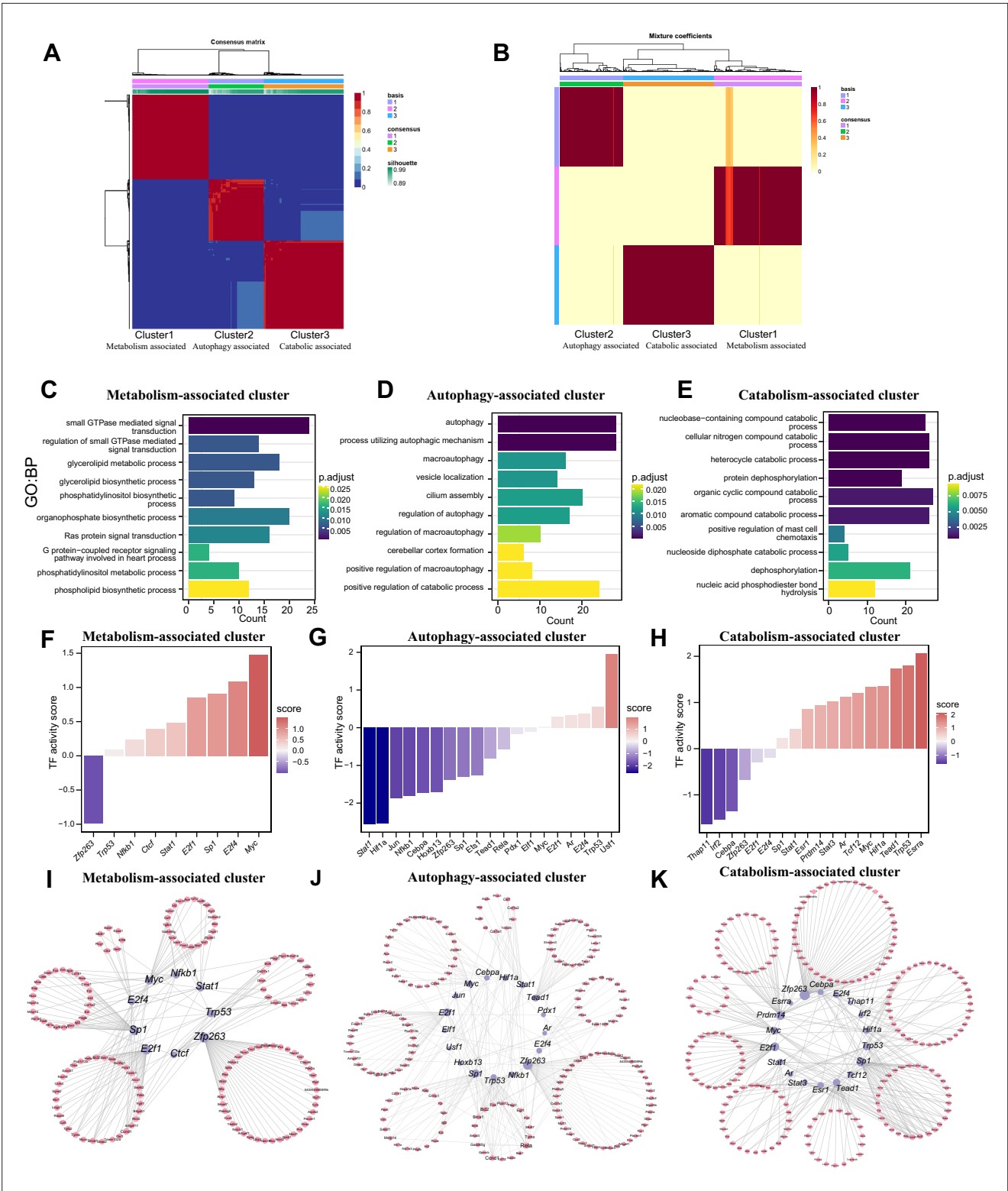

**Figure 6.** Three patterns regulate m¹A modification in neurons. (**A**) NMF analysis shows three m1A modification clusters (termed the metabolism-associated cluster (MAC), autophagy-associated cluster (AAC), and catabolic-associated cluster (CAC)). (**B**) Mixture coefficients matrix also shows that the three clusters have good discrimination. (**C-E**) GO enrichment analysis shows the different biological functions of each cluster. (**F-H**) Bar graph shows the transcription factors activity in each m¹A pattern. (**I-K**) TF-target genes regulatory networks in m1A patterns.

The online version of this article includes the following figure supplement(s) for figure 6:

**Figure supplement 1.** Three patterns regulate m¹A modification in neurons.

## Discussion

m[1]A is an abundant modification across eukaryotes. Although some studies have reported the landscape and characteristics of m[1]A modification in tissues and cells (*Dominissini et al., 2016*; *Roundtree et al., 2017*; *Safra et al., 2017*), its features and functions in neurons remain unclear. In this study, we identified m[1]A modifications on three kinds of RNAs (mRNAs, lncRNAs, and circRNAs) in normal neurons and OGD/R-treated neurons. We identified thousands of m[1]A mRNAs and found that the number and level of mRNA m[1]A modifications were increased after OGD/R. Regarding lncRNAs, in addition to exploring the basic features of m[1]A modification, we found two important m[1]A lncRNAs, *Meg3* and *Neat1*, might be associated with RBPs and possibly play vital roles in the regulation of mRNA metabolism. There are approximately 3000 m1A circRNAs in neurons, and they play roles in the nervous system. The m[1]A modifications on circRNAs such as *App* and *Foxo5* may be involved in the mechanism of circRNA translation. Furthermore, we explored the underlying associations among m[1]A RNAs (circRNAs, lncRNAs, and mRNAs) and found that there are two different complex regulatory networks for differential expression and differential methylation. We confirmed that m[1]A modification of some mRNAs affects the function of other regulatory mechanisms. The m[1]A modification patterns in neurons were also investigated. Three m[1]A patterns were identified: the Metabolism-associated cluster, Autophagy-associated cluster, and Catabolism-associated cluster. And there are specific TF networks in these three m[1]A patterns.

Relatively conserved sequences in RNAs are usually identified as sites of various chemical modifications. Methyltransferases can modify RNAs by recognizing those motifs. There are different motif sequences for different RNA modifications. m[6]A modification, the most important and common modification from yeast to humans, is located predominantly at RRACU (where *R*=A/G) consensus motifs in mammals and RGAC (where *R*=A/G) consensus motifs across yeast species (*Roundtree et al., 2017*). Several studies have identified motifs for m[1]A modification. The GUUCRA motif was identified in the mitochondrial transcriptome (*Han et al., 2017*). Another study also indicated that m[1]A was preferentially located at a GA-rich motif in 10–15% of cases (*Dominissini et al., 2016*). The motifs identified in these different studies were inconsistent with our findings. We identified different motifs in neurons, mRNAs (CMGCWGC and GCGGCGGCGGC), lncRNAs (AGARRAARAARAARA and TGCTGCTGCTG), and circRNAs (CWTCNTC and TGGARRA). Some modifications (*Dominissini et al., 2016*; *Safra et al., 2017*), such as m[1]A, are more likely to be present in regions with a high GC content, as found in previous studies. In addition, our results show that m[1]A-modified motifs may have RNA species specificity. These m[1]A motifs differ greatly across RNAs. Combining our findings with those of a previous study indicating that motifs have RNA methyltransferase specificity (*Zhang and Jia, 2018*), we speculate that different methyltransferases may regulate the modification of different RNAs. Currently, the results of motif identification seem relatively complex. Further identification of different methylases for different RNAs combined with experimental verification may be a future research direction.

It has been shown that m[6]A modifications are time-specific in developmental processes as well as in the development of some diseases. During both *Drosophila* development and mouse cerebellum development (*Lence et al., 2016*; *Ma et al., 2018*), the level of m[6]A modification changes dynamically with the developmental process, which is associated with various organogenesis and specific functions. In Alzheimer's disease, m[6]A modification levels also change as the disease progresses (*Shafik et al., 2021*). However, whether the m[1]A modification is time-specific is still unknown. In the present study, we focused on the changes in m[1]A modification under different OGD/R treatments. Our results suggested that m[1]A regulators differentially expressed and mRNA and noncoding RNA has their own features in different OGD/R treatments. We suggested that this is partly indicative of the time-specificity of m[1]A modifications under different OGDR treatments, but more investigations are needed to determine whether m[1]A modifications are time-specific during development and in other disease progressions.

Different RNAs play different roles in biological processes. mRNAs are translated into proteins that regulate biological activities, while noncoding RNAs, such as lncRNAs and circRNAs, play important roles in posttranscriptional regulation via various mechanisms. Chemical modification of these RNAs changes their original regulatory mechanisms and influences their downstream effects. m[6]A modification of mRNAs can affect mRNA expression, stability, and splicing and can promote or prevent mRNA degradation (*Tang et al., 2018*; *Song et al., 2020*). m[6]A modification of circRNAs can affect the translation of circRNAs and splicing of longer circRNAs (*Wesselhoeft et al., 2019*; *Huang et al.,*

*2021*). Similarly, m⁶A modification of lncRNAs also plays vital roles in lncRNA-mediated gene silencing and RNA stability (*Zheng et al., 2019*; *Yang et al., 2020b*). Currently, few studies have addressed the effect of m¹A modification on RNA metabolism and functions. Some studies have shown that m¹A modification of mRNA can regulate mRNA translation. We found that m¹A modification can affect the ceRNA mechanism by affecting the binding of miRNAs to mRNAs. Similarly, little is known about the function of m¹A modification on noncoding RNAs. In this study, we identified the features of m¹A modification on lncRNAs and circRNAs and summarized the numbers of genomic sources and intergroup differences in m1A RNAs. Regarding m¹A lncRNAs, we explored the RBPs that may interact with m¹A lncRNAs and found that the RBPs bound to m¹A lncRNAs are involved mainly in the metabolism and splicing of mRNAs. Regarding m¹A circRNAs, in addition to providing a basic landscape, we also studied the translation characteristics of m1A circRNAs. We found that m¹A circRNAs might be translated into proteins and that some of these m¹A circRNAs also have m⁶A-modified sites. Other studies *Huang et al., 2021* have shown that m⁶A modification also regulates circRNA translation. Therefore, if a circRNA simultaneously contains two modifications, which modification has a greater impact on translation regulation, and what is the underlying mechanism? These questions deserve in-depth study. A complex relationship may exist between m⁶A modifications and m¹A modifications. m¹A can be converted to m⁶A under heat and alkaline conditions, which is known as Dimroth rearrangement (*Liu et al., 2022*). m¹A modification has also been found to facilitate m⁶A-mediated mRNA degradation via HRSP12 (m¹A 'reader')-YTHDF2 (m⁶A 'reader'), which provides evidence for a crosstalk between different RNA modifications (*Boo et al., 2022*). We speculate that there may also be interactions between different modifications in neurons, which is an interesting and meaningful direction worthy of further exploration.

Transcriptome-wide m¹A modification may play a role in some currently identified posttranscriptional regulatory mechanisms. The regulatory mechanism of ceRNAs is generally considered to play a role at the expression level, but the effect of m1A modifications on this mechanism is not clear. In this study, two interaction networks were constructed by using differential expression and differential methylation data, and both networks were extremely complex. Similar regulatory loops between lncRNAs and circRNAs were formed by linking miRNAs, especially in the differential modification regulatory network. Modifications of these RNAs may affect the interactions in these loops. For example, m¹A modification of lncRNAs, circRNAs and mRNAs may affect their interactions with miRNAs. This mechanism provides a more accurate strategy for the upstream and downstream regulation of miRNAs, which in turn more conveniently regulates biological processes. In this study, we also found multiple miRNA−mRNA pairs, and the dual luciferase assay confirmed that m¹A modification of some mRNA 3'UTRs affects the binding of miRNAs to these mRNAs. We found that these genes affected by m¹A modification (*Rbfox3* and *Arid1a*) are related to neural development and differentiation; the genes that were not affected (*Grin2d* are *Wdtc1*) are related mostly to broader cellular functions. Therefore, we asked whether m¹A modification preferentially occurs on certain genes. m¹A modification depends on several methyltransferases. Thus, do methyltransferases exhibit preferences for different functional genes? Just as we proposed that m¹A modification of different RNA species may be mediated by different m¹A-related enzymes, we hypothesize that different methyltransferases may exhibit preferences for different functional genes.

The m¹A modification patterns remain unclear in neurons. As our results showed, m¹A modification of different RNAs may be regulated by different enzymes, and different functional genes may be regulated via different methyltransferases; thus, identification of m¹A patterns is urgently needed. We identified three patterns of m¹A modification in neurons: a metabolism-associated cluster, an autophagy-associated cluster, and a catabolism-associated cluster. The genes with each pattern of m¹A modification perform different biological functions. These three m¹A patterns are closely related to the fate of neurons. The proper functioning of neuronal metabolic mechanisms ensures the function of neurons (*Dienel, 2019*). The metabolic pathways of these neurons may be regulated by m¹A modifications. Disruption of these metabolic pathways may contribute to the initiation or progression of neurological diseases (*Bonvento and Bolaños, 2021*). Autophagy is also important in neurons. Autophagy pathways are essential in neurodevelopment as well as in the maintenance of neuronal homeostasis (*Stavoe and Holzbaur, 2019*). An imbalance in catabolic processes can also lead to neuronal damage (*Camandola and Mattson, 2017*; *Ravera et al., 2019*). Based on previous studies in other cells (*Dominissini et al., 2016*; *Safra et al., 2017*; *Zhao et al., 2019*), we found that the function

of the m$^1$A in different cell types seems to be different and we speculate that these m$^1$A modifications patterns are cell-type-specific. Investigation of this possibility also requires more sequencing and experimental data for analysis.

With the deepening of the RNA epitranscriptomic research studies, some m$^6$A regulators have been selected as targets for the development of corresponding drugs, especially within the field of oncology. Nucleoside and non-nucleoside METTL3 inhibitors were designed to inhibit the function of METTL3 in different kinds of tumours and proved METTL3 was an efficient therapeutic target (*Xu and Ge, 2022*). METTL14 also plays an important role in a variety of tumours. Designing activators and inhibitors of METTL14 is also an important direction for drug development (*Guan et al., 2022*). In addition, m$^6$A-modified noncoding RNAs are potential therapeutic targets (*Yi et al., 2020*). In our current research, we have not only identified differentially expressed m$^1$A regulators (*Alkbh3*, *Trmt10c*, *Trmt61a*, *Ythdf2*, and *Ythdf3*) in OGD/R, but also constructed noncoding RNA regulatory networks for differential expression and differential methylation under OGD/R treatment, and identified a number of noncoding RNAs that may play important roles, all of which may be targets for mitigating the effects of OGD/R.

Overall, in this study, we profiled the features and patterns of m$^1$A modification in neurons and OGD/R-treated neurons. We identified m$^1$A modifications on different RNAs and explored the possible effects of m$^1$A modification on the functions of different RNAs and the posttranscriptional regulation mechanism. Although we found and proposed some roles related to m$^1$A modification, it is still a relatively complex modification type (no completely conserved modification motif has been found, and m$^1$A may exhibit RNA type specificity, cell type specificity, etc.). Therefore, as more research techniques are developed (*Xie et al., 2021*), in-depth study of the m$^1$A modification for various biological processes and the development of related therapeutic targets based on this modification require considerably integrated bioinformatic and basic experimental research studies.

## Materials and methods

### Animals

C57BL/6 mice purchased from the Laboratory Animal Centre, Academy of Military Medical Science (Beijing, China), were used mainly for harvesting primary cerebral neurons. All experiments were performed in adherence to the National Institutes of Health Guidelines for the Care and Use of Laboratory Animals and were approved by the Medical Ethics Committee of Qilu Hospital of Shandong University (IACUC Issue NO. DWLL-20210061).

### Primary cerebral neuron isolation and culture

Primary cultures of cerebral neurons were obtained as described previously (*Hilgenberg and Smith, 2007*). In brief, mouse foetuses (embryonic day 17 [E17]) were removed from the uterus, and the individual foetuses were freed from the embryonic sacs. The brain and cortical tissue were dissected and placed in high-glucose Dulbecco's modified Eagle's medium (DMEM-HG) without phenol red (Gibco, Grand Island, NY, USA; Cat. No. 31053028). Papain solution (10 U/mL; Sigma−Aldrich, St. Louis, MO, USA; Cat. No. LS003126) was added to these cerebral tissues, which were then incubated for 15 min at 37°C in a 5% CO$_2$ incubator. Dissociated cortical cells were plated on poly-L-lysine (Sigma−Aldrich; Cat. No. P4832)-coated cell culture dishes and cultured in DMEM-HG (Gibco, Grand Island, NY, USA; Cat. No. 31053028) containing 10% foetal bovine serum (Gibco, Australia; Cat. No. 10099141) and 1% penicillin/streptomycin (P/S; Invitrogen, Carlsbad, CA, USA; Cat. No. 15140148) at a density of 1.0×10$^6$ cells/mL. Four hours after seeding, the medium was replaced with neurobasal medium (NM; Gibco, Carlsbad; Cat. No. 21103049) supplemented with B-27 (Gibco, Grand Island, NY, USA; Cat. No. 17504044). Cells were cultured in a humidified incubator at 37°C with 5% CO$_2$. The medium was changed every 3 days. Cultures were used for in vitro experiments after 7 days.

### OGD/R modelling

The OGD/R model was established using a previously described method with slight modifications (*Tasca et al., 2015*; *Ryou and Mallet, 2018*). Cultured primary cerebral neurons were washed twice with phosphate-buffered saline (PBS; Sigma−Aldrich; Cat. No. D8537) supplemented with 1% P/S after 7 days of culture. Glucose-free Dulbecco's modified Eagle's medium (Gibco, Grand Island, NY,

USA; Cat. No. 31053028) was added to the dishes. Next, the neurons were cultured with a GENbag anaerobic incubation system (bioMérieux SA, France; Cat. No. 45534) at 37°C. The cultures were kept separate under hypoxic conditions for 1.5 hr and 3 hr to achieve OGD. Thereafter, the neurons were allowed to recover by culture in a normal serum-free medium (NM) under normal incubation conditions (37°C, 5% $CO_2$) for 24 hr. Neurons cultured in a normal serum-free medium under normoxic conditions served as Controls.

### RNA isolation
Total RNA was extracted from primary cultured neurons using TRIzol reagent (Invitrogen, Carlsbad, CA, USA; Cat. No. 15596018) according to the manufacturer's protocol. A NanoDrop ND-1000 system (Thermo Fisher Scientific, Waltham, MA, USA) was used to measure the RNA concentration in each sample. The OD260/OD280 ratio was assessed as an index of RNA purity, and samples with OD260/OD280 values ranging from 1.8 to 2.1 met the qualifications for purity. RNA integrity was evaluated using denaturing agarose gel electrophoresis.

### Preparation of the m$^1$A RNA immunoprecipitation sequencing (MeRIP-seq) library
MeRIP-seq was performed by Cloudseq Biotech Inc (Shanghai, China) according to a published procedure with slight modifications (*Meyer et al., 2012*). In brief, three biological replicates were used for the Control, OGD/R 1.5 hr, and OGD/R 3 hr groups. Isolated RNA was chemically degraded into fragments of approximately 100 nucleotides in length using a fragmentation buffer (Illumina, Inc, CA, USA). GenSeq m1A MeRIP Kit (Cloudseq Biotech Inc, Shanghai, China; Cat. No. GS-ET-002) was used to perform immunoprecipitation (IP) of m1A RNA according to the manufacturer's recommendations. IP buffer, Protein A/G beads, and m1A antibodies were used to prepare the magnetic beads for IP. Next, the MeRIP reaction mixture was prepared according to the manufacturer's guidelines, and fragmented RNA was included in this mixture. The magnetic beads and MeRIP reaction mixture were combined in tubes, and all tubes were incubated with rotation for 2 hr at 4°C. Subsequently, elution buffer was prepared according to the manufacturer's instructions and was used to elute bound RNA from the beads using the anti-m$^1$A antibody in IP buffer. Both input samples without IP and m$^1$A input samples were used for library construction using a NEBNext Ultra II Directional RNA Library Prep Kit (New England Biolabs, Inc, MA, USA). Eluted RNA fragments were converted to cDNA and sequenced or treated to induce partial m$^1$A to m$^6$A conversion before cDNA synthesis. Library sequencing was performed using an Illumina HiSeq 4000 instrument (Illumina, Inc, CA, USA) with 150 bp paired-end reads.

### MeRIP-seq data analysis
In brief, quality control of the paired-end reads was performed with FastQC (v0.11.9) prior to trimming of 3' adaptors and removal of low-quality reads using Cutadapt software (v1.9.3). Then, HISAT2 software (v2.0.4) was used to align the clean reads from all libraries to the reference genome (mm10) downloaded from Ensembl. Peaks for which −10×log10(*p* value)>3 were detected using Model-Based Analysis of ChIP-Seq (MACS) software (v2.2.7.1). Differentially methylated sites with a fold change of ≥2 and false discovery rate (FDR) of ≤0.0001 were identified with the diffReps differential analysis package (v1.55.6). The peaks identified by MACS and diffReps that overlapped with exons of mRNAs, lncRNAs, and circRNAs were identified and selected for further analysis. In addition to conventional peak calling (conventional method), the peaks with A->T mismatches (mismatch method) were analysed. In addition, m$^1$A sites have a partial chance of terminating reverse transcription, resulting in lower coverage in the middle of the peak (near the m$^1$A site) than on both sides, forming a depression. These peaks and depressions (trough method) were also analysed. Moreover, when m$^1$A was converted into m$^6$A, the mismatch and termination properties of m$^6$A were lost, while reverse transcription was normal. Thus, after sequencing, peaks with the normal shape could be detected at the sites (treatment method).

### Preparation of RNA sequencing (RNA-seq) libraries
Total RNA was used for the removal of ribosomal RNA (rRNA) using Ribo-Zero rRNA Removal Kits (Illumina, USA) following the manufacturer's instructions. RNA libraries were constructed by using

rRNA-depleted RNAs with a TruSeq Stranded Total RNA Library Prep Kit (Illumina, USA) according to the manufacturer's instructions. Libraries were subjected to quality control and quantified using a BioAnalyzer 2100 system (Agilent Technologies, USA). RNA was purified from each sample using Oligo(dT) Dynabeads (Invitrogen) and subjected to first-strand cDNA synthesis and library preparation using a TruSeq Stranded mRNA Library Prep Kit (Illumina). Libraries (10 pM) were denatured as single-stranded DNA molecules, captured on Illumina flow cells, amplified in situ as clusters, and finally sequenced for 150 cycles on an Illumina HiSeq 4000 instrument according to the manufacturer's instructions.

### RNA-seq data analysis

Quality control of paired-end reads was performed with FastQC (v0.11.9) prior to trimming of 3' adaptors and removal of low-quality reads using Cutadapt software (v1.9.3). The high-quality reads were aligned to the mouse reference genome (mm10) with HISAT2 software (v2.0.4). Then, guided by the Ensembl (GRCm39.104) GTF gene annotation file, expression was estimated in units of fragments per kilobase of transcript per million mapped reads (FPKMs). The differentially expressed genes were identified as those with a fold change of ≥2 and adjusted p value of ≤0.05.

### Preparation of miRNA sequencing (miRNA-seq) libraries and data analysis

miRNA-seq was conducted at Cloudseq Biotech Inc (Shanghai, China). In brief, total RNA from each group was prepared and quantified with the BioAnalyzer 2100 system (Agilent Technologies, USA). Polyacrylamide gel electrophoresis (PAGE) was performed, and the gel was cut to select the band corresponding to a length of 18–30 nt to recover small RNAs. Adaptors were then ligated to the 5' and 3' ends of the small RNAs. After cDNA synthesis and amplification, the PCR-amplified fragments were purified from the PAGE gel, and the complete cDNA libraries were quantified with a BioAnalyzer 2100. Sequencing was performed on the Illumina HiSeq 4000 instrument, and 50 bp single-end reads were generated. For miRNA-seq data analysis, the adaptor sequences were trimmed, and the trimmed reads (≥15 nt) were retained by Cutadapt software (v1.9.3). Then, the trimmed reads were aligned to the merged mouse pre-miRNA databases (known pre-miRNAs from miRbase [v22.1]) using NovoAlign software (v3.02.12). The number of mature miRNA-mapped tags was defined as the raw expression level of that miRNA. Read counts were normalized by tag counts per million aligned miRNAs (TPM) values. Differentially expressed miRNAs between the two groups were filtered by the following criteria: fold change ≥2 and adjusted *p-value* ≤0.05.

### Functional annotation

Gene Ontology (GO) and Kyoto Encyclopedia of Genes and Genomes (KEGG) pathway enrichment analyses based on the differentially expressed genes were performed with the R package 'clusterProfiler' (v4.4.0) (*Wu et al., 2021*). GO covers three categories: cellular component (CC), molecular function (MF), and biological process (BP). The adjusted *p* value for a GO term denotes the significance of the enrichment of genes in that term. Pathway enrichment analysis is a functional analysis that maps genes to KEGG pathways. The Fisher p value denotes the significance of the pathway correlation to the conditions. GO analysis was performed using *enrichGO* function in the R package 'clusterProfiler' with the following parameters: pvalueCutoff = 0.005, qvalueCutoff = 0.005, minGSSize = 2, and maxGSSize = 500. KEGG analysis was performed using *enrichKEGG* function in 'clusterProfiler' with the following parameters: pvalueCutoff = 0.05 and qvalueCutoff = 0.05. The GO and KEGG pathway terms with adjusted *p* values of ≤0.05 were considered significantly enriched.

### lncRNA–RBP interaction networks construction

Based on differentially methylated lncRNA profile (*Figure 3—source data 1*), we predicted RBPs that might interact with those lncRNA by the ENCORI database (https://starbase.sysu.edu.cn/). We predicted the lncRNA–RBP interaction network using the RBP-Target module in ENCORI with the following parameters: Clade = mammal, Genome = mouse, CLIP Data = medium stringency ≥ 2. We then integrated the lncRNA–RBP interacting pairs using R tidyverse package (v 1.3.2). Network visualization was done using Cytoscape vs 3.9.1 (http://www.cytoscape.org/). The shape and colour of the points are customizable.

## circRNA structure prediction and protein structure prediction of circRNA-encoded peptides

circRNA structure and polypeptide structure were all predicted by the riboCIRC database (*Li et al., 2021*) (http://www.ribocirc.com/index.html). For the circRNA structure prediction, after submitting the query, the site provides several pieces of information, including ribosome associated evidence, $m^6A$ evidence, and ORF evidence. We redrew the RNA structure from the website using Adobe Illustrator CC 2017 and added additional information. For the protein structure prediction, the I-TASSER Suite was used (*Yang et al., 2015*). After submitting the query, predicted secondary structure, predicted solvent accessibility, top 10 threading templates used by I-TASSER and top 5 final models predicted by I-TASSER were provided by the website tool.

## Competing endogenous RNA (ceRNA) regulatory network construction

Based on the miRNA expression data (*Figure 5—source data 1*), we used three databases (TargetScan, miRDB, and miRTarBase) for comprehensive analysis of miRNA binding target genes. We obtained the intersection of the mRNAs predicted by the three databases, and differentially expressed mRNAs and differentially methylated mRNAs were integrated to construct the miRNA–mRNA interaction network. For lncRNAs and circRNAs, we used the ENCORI database (https://starbase.sysu.edu.cn/) to analyse the binding of lncRNAs and circRNAs to the differential miRNAs. After obtaining the lncRNA–miRNA and circRNA–miRNA interaction networks, we intersected the predicted lncRNAs and circRNAs with the differentially expressed lncRNAs and circRNAs and the differentially methylated lncRNAs and circRNAs in our data. Finally, we integrated all data to construct the expression and methylated circRNA/lncRNA–miRNA–mRNA interaction networks. Network visualization was done using Cytoscape vs 3.9.1 (http://www.cytoscape.org/). The shape and colour of the points are customizable.

## Luciferase activity assay

The Rbfox3, Arid1a, Grin2d, and Wdtc1 reporter vectors were constructed by inserting the 3'UTRs of the corresponding genes downstream of the luciferase reporter gene into the GV272 plasmid (GeneChem Technologies, Shanghai, China). The Trmt6 overexpression vector was constructed by subcloning the coding sequence (CDS) into the GV712 plasmid (GeneChem Technologies, Shanghai, China). The *Mir129*, *Mir101b*, *Mir27*, and Mir*16* mimics were designed and synthesized by GeneChem. We used Lipofectamine 3000 to cotransfect reporter plasmids, the Trmt6 plasmid, and miRNA mimics into HEK293T cells. Luciferase activity was measured 48 hr later according to the manufacturer's procedures (E292, Promega, USA). All experiments described were replicated independently with similar results at least three times.

## Nonnegative matrix factorization (NMF) clustering

NMF is a widely used tool for the analysis of high-dimensional data because it automatically extracts sparse and meaningful features from a set of nonnegative data vectors. NMF clustering was used to determine the $m^1A$ modification patterns in our nine differently treated samples (*Gaujoux and Seoighe, 2010*). We performed this analysis using all $m^1A$-modified mRNAs in the three groups. The *k* value at which the magnitude of the cophenetic correlation coefficient began to decrease was selected as the optimal number of clusters. The heatmap of $m^1A$ regulators, the basic components, and the NMF connectivity matrix for different clusters were estimated with the NMF package (v0.24.0) in R (v4.1.3).

## Transcription factor (TF) analysis and TF-target genes network construction

TF analysis was employed to determine the upstream regulation of three $m^1A$ patterns. Submit the genes contained in each of the three patterns to the 'Enrichr' database (http://amp.pharm.mssm.edu/Enrichr/) and select the ChEA 2022 database to view the results of the enriched TFs. To predict the activity of TFs in each pattern, we used the R package 'decoupleR' (v 2.5.0). The data containing signed tTF - target gene interactions were harvested from the R package 'decoupleR' (v 1.10.0). We employed the *run_wmean* function to calculate the TF activity with the following parameters: mor='mor', times = 1000, and minsize = 5. Network visualization was done using Cytoscape vs 3.9.1. The shape and colour of the points are customizable.

## Pathway activity and correlation analysis

The R package 'GSVA' (v 1.46.0) was exploited to quantify the pathway activity in three $m^1A$ patterns by calculating the GSVA score. Genes for each pathway were obtained from the corresponding GO database (http://geneontology.org/). The *gsva* function was used to calculate the pathway score. Then we used the *corr.test* function from R package 'psych' (v 2.2.9) calculated the correlation between the $m^1A$_regulators and pathway score.

## Acknowledgements

We also thank Dr Jianming Zeng (University of Macau), and all the members of his bioinformatics team, biotrainee, for generously sharing their experience and codes. Funding This study was funded by the National Natural Science Foundation of China (81972073), Taishan Scholars Program of Shandong Province-Young Taishan Scholars (tsqn201909197), and National Key Research and Development Project of Stem Cell and Transformation Research (2019YFA0112100).

## Additional information

### Funding

| Funder | Grant reference number | Author |
|---|---|---|
| National Natural Science Foundation of China | 81972073 | Hengxing Zhou |
| Taishan Scholars Program of Shandong Province-Young Taishan Scholars | tsqn201909197 | Hengxing Zhou |
| National Key Research and Development Program of China Stem Cell and Translational Research | 2019YFA0112100 | Shiqing Feng |

The funders had no role in study design, data collection and interpretation, or the decision to submit the work for publication.

### Author contributions

Chi Zhang, Data curation, Formal analysis, Visualization, Writing – original draft; Xianfu Yi, Conceptualization, Formal analysis; Mengfan Hou, Formal analysis, Visualization; Qingyang Li, Xueying Li, Methodology; Lu Lu, Enlin Qi, Mingxin Wu, Data curation, Methodology; Lin Qi, Huan Jian, Formal analysis, Investigation; Zhangyang Qi, Validation; Yigang Lv, Validation, Methodology; Xiaohong Kong, Investigation, Methodology; Mingjun Bi, Conceptualization, Writing – review and editing; Shiqing Feng, Hengxing Zhou, Conceptualization, Supervision, Funding acquisition, Writing – review and editing

### Author ORCIDs

Chi Zhang http://orcid.org/0000-0003-1288-9006
Mingjun Bi http://orcid.org/0000-0001-8748-2636
Hengxing Zhou http://orcid.org/0000-0003-0053-8970

### Ethics

Animal experimentation: All experiments were performed in adherence to the National Institutes of Health Guidelines for the Care and Use of Laboratory Animals and were approved by the Medical Ethics Committee of Qilu Hospital of Shandong University (IACUC Issue NO. DWLL-20210061).

### Decision letter and Author response

Decision letter https://doi.org/10.7554/eLife.85324.sa1
Author response https://doi.org/10.7554/eLife.85324.sa2

## Additional files

### Supplementary files
• MDAR checklist

### Data availability

Sequencing data have been deposited in the BIGD (National Genomics Data Center, Beijing Institute of Genomics, Chinese Academy of Sciences) database under the bioproject accession code PRJCA010978. All data generated or analysed during this study are included in the manuscript and supporting file; Source Data files have been provided for Figures 2, 3, 4 and 5.

The following dataset was generated:

| Author(s) | Year | Dataset title | Dataset URL | Database and Identifier |
|---|---|---|---|---|
| Zhang C, Yi X, Hou M | 2023 | The landscape of m1A modification and its post transcriptional regulation functions in primary neuron | https://ngdc.cncb.ac.cn/bioproject/browse/PRJCA010978 | BioProject, PRJCA010978 |

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
