## [Editor Report]

This study presents a rather valuable finding on the exploration of RNAs m1A modification in normal and OGD/R-treated neurons and the effects of m1A on diverse RNAs. The evidence supporting the claims of the authors is quite solid. The work will be of interest to scientists working in the field of m1A modifications. Noteworthy, this manuscript provides a new avenue for the development of novel therapeutics against ODG/R-related disease.

---

## [Decision Letter]

**Decision letter after peer review:**

Thank you for submitting your article "The landscape of m1A modification and its posttranscriptional regulatory functions in primary neurons" for consideration by *eLife*. Your article has been reviewed by 3 peer reviewers, one of whom is a member of our Board of Reviewing Editors, and the evaluation has been overseen by Caigang Liu as the Senior Editor. The reviewers have opted to remain anonymous.

Essential revisions:

1) The authors should provide all the technique details for the bioinformatics analysis presented in the manuscript.

2) The authors should conduct in-depth literature search about the m1A regulators and perform gene expression profile analyses to identify differentially expressed regulators.

3) The authors should explore the role of m1A regulators and, in particular, the association with the three m1A modification patterns.

4) In Figure 6, the authors presented three patterns regulate m1A modification in neurons. The authors should further analyze the transcription factors to clarify the upstream regulation of these three patterns.

*Reviewer #1 (Recommendations for the authors):*

Some specific points for the manuscript are as follows,

1. The authors should provide all the technique details for the bioinformatics analysis presented in the manuscript. This is important because the reader should be able to reproduce all the results following their technique protocols. For instance, Figure 2D-E-F, Figure 3E-F, Figure 4E-F, Figure 5A-B-C-D-E and etc. This is mandatory.

2. Technique details for the predicted polypeptide structures in Figure 4H and Figure S4H should also be provided.

3. The authors claimed the identification of three patterns of m1A modifications. What are the underlying functional links of these patterns with neurons fate? The authors should discuss this in length in the manuscript.

4. Have the authors examined the m1A modifications of tRNA in neurons and OGD/R-treated neurons?

5. Have the authors examined the topology of the m1A modification in normal and OGD/R-treated conditions or on diverse RNAs? This should be discussed.

6. In page 3, line 46, the authors claimed that the present work provide "a theoretical basis for treating and developing drugs for OGD/R pathology-related diseases". The authors should elaborate this point in the Discussion section.

7. Language inconsistences should be fixed throughout the manuscript. This is mandatory.

*Reviewer #2 (Recommendations for the authors):*

These findings deepened the understanding of m1A modification in neuron and provided new insights into the pathogenic mechanism and pathological process of OGD/R. In general, this study is innovative, practical and well designed. Despite the interesting results obtained, there are some concerns and criticisms with their observations/conclusions.

1. Despite the authors' comprehensive exploration of m1A modifications in neuron, they seem to have overlooked an important aspect: are m1A regulators differentially expressed in normal neurons and OGD/R neurons? The currently known m1A regulators include: Trmt6, Trmt61A, Alkbh1, Alkbh3, Fto, Ythdf1, Ythdf2 and Ythdc1. The authors could also do more in-depth literature search about the m1A regulators and performed gene expression profile analyses to identify differentially expressed regulators.

2. Considering the research frontier, the biological role of m1A methylation modification should be described in detail in the Introduction section. M1A modification has been found to play a critical role in various biological conditions and diseases. Recent studies have also shown that m1A methylation regulates glycolysis of cancer cells through modulating ATP5D (PMID: 35867754, 2022) and also participate in the modulating of macrophage polarization which promotes aortic inflammation (PMID: 35620523, 2022), authors should review the latest literature on m1A modification and describe the role and significance of m1A modification in depth.

3. In Figure 6, the authors presented three patterns regulate m1A modification in neurons and the functions of these three patterns are identified. I consider this result to be of great interest. I suggest that the authors could further analyze the transcription factors to clarify the upstream regulation of these three patterns. Transcription factor analysis can tell us at least two things: 1. whether there is specific transcription factor expression in different patterns; 2. whether there is a regulatory network of transcription factors in different patterns.

*Reviewer #3 (Recommendations for the authors):*

Overall, this is an interesting and well performed study that described a comprehensive landscape of m1A modification in primary neuron and investigated the role of m1A in the circRNA/lncRNA‒miRNA-mRNA regulatory network following OGD/R. The focus on the two different complex regulatory networks for differential expression and differential methylation is important and it will be a valuable resource for the research community that focuses on epitranscriptomics and central nerve system diseases. Collectively, the authors present an exciting piece of work that certainly adds to the literature regarding epitranscriptomic features in neuron. While interesting results obtained and the paper is nicely written, I have several comments the authors might want to consider to improve the overall strength of their manuscript.

1. In this manuscript, a meaningful conclusion is that the m1A modification on mRNA 3'UTR hinders the binding of miRNAs (Figure 5). I think this provides a good idea to study the possible roles of m1A modification in a variety of diseases. I would like to know what the authors envision for the continuation of research in this area.

2. The authors present three patterns ("metabolism-associated cluster", "autophagy- associated cluster", "catabolism-associated cluster") of m1A modifications in the last part of RESULTS (Figure 6), which is an intriguing finding. However, it does not seem to explore the specific details of these three modification patterns further. I suggest that the authors could further analyze whether the genes in these patterns are regulated by some common transcription factors.

3. The m1A modification mainly depends on the dynamic regulation of RNA methyltransferases (writers), demethylases (erasers) and m1A-binding proteins (readers). However, in this manuscript the authors do not mention the changes in m1A regulators. I am curious about whether m1A regulators will be differentially expressed after OGD/R and suggest the authors to add this part of the analysis. Of course, if differentially expressed m1A regulators exist, the authors could further explore the role of these factors and, in particular, the association with the three m1A modification patterns.

4. The authors obtained interesting results at the two time points set in the OGD/R model, they should try to explain the possible effects of such changes on neuronal biological processes in Discussion.

5. Figure 3D and Figure 4D: The title of the Y-axis might be mislabeled.

6. Figure 4 and Figure 6 has an additional caption at the bottom of the image, please double check.

7. P28 Line547 and P30 L575: The abbreviations of reference genome are inconsistent.

---

## [Author Response]

Essential revisions:1) The authors should provide all the technique details for the bioinformatics analysis presented in the manuscript.

Special thanks to Editors for the pertinent suggestions. We have elaborated on the technique details for the bioinformatics analysis in the Materials and methods section.

2) The authors should conduct in-depth literature search about the m1A regulators and perform gene expression profile analyses to identify differentially expressed regulators.

This is an excellent point that is brought forth by the Editors and Reviewers. we have performed an additional analysis to identify the differentially expression of m1A regulators in OGD/R-treated neurons. The results showed that the expression of Alkbh3, Trmt10c, Trmt61a, Ythdf2, and Ythdf3 were statistically significant (Figure 1F). However, the trends of these m1A regulators were different under different OGD/R treatments. This new data analysis now appears in Figure 1F of the revised manuscript.

3) The authors should explore the role of m1A regulators and, in particular, the association with the three m1A modification patterns.

We appreciate the insightful suggestions and agree with the reviewer. To explore the relationship between m1A regulators and three m1A modification patterns, we used GSVA for scoring the major pathways in the three patterns and subsequently calculated the correlation between these m1A regulators expression and the 3 pattern pathway scores. Our results suggested that most of m1A regulators were positively correlated with metabolism-associated functions and autophagy-associated functions, and negatively correlated with catabolism-associated functions (Figure 6—figure supplement 1H). We have added this part of content in the Figure 6—figure supplement 1 of the revised manuscript.

4) In Figure 6, the authors presented three patterns regulate m1A modification in neurons. The authors should further analyze the transcription factors to clarify the upstream regulation of these three patterns.

We thank the reviewer for this helpful comment. To clarify the transcription factor regulation of different patterns, we have performed new analyses to identify activity of transcriptional factors in those three patterns. We found that the representative transcription factors enriched specifically in cluster. Further prediction of the activity of these transcription factors showed that different transcription factor activation and repression patterns exist in different clusters. In MAC and CAC, most of the transcription factors had increased inferred activity. However, most of the transcription factors had decreased inferred activity in AAC (Figure 6F-6H). Based on these transcription factors with altered activity, we constructed a TF-target gene regulatory network (Figure 6I-6K). This new data analysis now appears in Figure 6 of the revised manuscript.

Reviewer #1 (Recommendations for the authors):Some specific points for the manuscript are as follows,1. The authors should provide all the technique details for the bioinformatics analysis presented in the manuscript. This is important because the reader should be able to reproduce all the results following their technique protocols. For instance, Figure 2D-E-F, Figure 3E-F, Figure 4E-F, Figure 5A-B-C-D-E and etc. This is mandatory.

Special thanks to you for your suggestions. We apologize for not presenting technique details clearly. Relevant detail protocols have added to the Materials and methods section.

2. Technique details for the predicted polypeptide structures in Figure 4H and Figure S4H should also be provided.

We agree with the reviewer and the protocols about how to predicted polypeptide structures have added to the Materials and methods section (circRNA structure prediction and protein structure prediction of circRNA-encoded peptides).

3. The authors claimed the identification of three patterns of m1A modifications. What are the underlying functional links of these patterns with neurons fate? The authors should discuss this in length in the manuscript.

Thank you very much for the constructive comments. After reviewing the literature, we meticulously discussed the relationship between these three patterns and neurons fate in the Discussion section.

4. Have the authors examined the m1A modifications of tRNA in neurons and OGD/R-treated neurons?

We appreciate the suggestion. We did not explore m1A modifications on tRNA in this manuscript. Here we just focused on three RNA types, mRNA, lncRNA, and circRNA. tRNA was beyond the scope of this study. However, this comment suggests a good direction for our future studies.

5. Have the authors examined the topology of the m1A modification in normal and OGD/R-treated conditions or on diverse RNAs? This should be discussed.

We are very sorry that the topology of the m1A modification was not examined in this manuscript. The main purpose of this manuscript is to construct a comprehensive m1A modification atlas in normal neurons and OGD/R-treated neurons. The m1A topology is achieved by two enzyme systems: m1A methyltransferase and m1A demethylases. We added the expression analysis of these enzymes in the in Figure 1F of the revised manuscript. In-depth study of m1A topology is an important direction for our future research.

6. In page 3, line 46, the authors claimed that the present work provide "a theoretical basis for treating and developing drugs for OGD/R pathology-related diseases". The authors should elaborate this point in the Discussion section.

We thank the Reviewer for this advice. Clarification of the mechanism of m1A modification in OGD/R may provide some reference for the treatment of this pathological process. We provide an additional description in the Discussion section:

“With the deepening of the RNA epitranscriptomic research studies, some m6A regulators have been selected as targets for the development of corresponding drugs, especially within the field of oncology. Nucleoside and non- nucleoside METTL3 inhibitors were designed to inhibit the function of METTL3 in different kinds of tumors and proved METTL3 was an efficient therapeutic target (Xu and Ge, 2022). METTL14 also plays an important role in a variety of tumors. Designing activators and inhibitors of METTL14 is also an important direction for drug development (Guan et al., 2022). In addition, m6A-modified noncoding RNAs are potential therapeutic targets (Yi et al., 2020). In our current research, we have not only identified differentially expressed m1A regulators (Alkbh3, Trmt10c, Trmt61a, Ythdf2, and Ythdf3) in OGD/R, but also constructed noncoding RNA regulatory networks for differential expression and differential methylation under OGDR treatment, and identified a number of noncoding RNAs that may play important roles, all of which may be targets for mitigating the effects of OGD/R.”

7. Language inconsistences should be fixed throughout the manuscript. This is mandatory.

We apologize for the inconsistent descriptions. We have read through carefully and changed the inconsistent contents. we have discussed the writing problems with a professional English supervisor, he gave us several suggestions, which have all been adopted and accordingly adjusted in the manuscript.

Reviewer #2 (Recommendations for the authors):These findings deepened the understanding of m1A modification in neuron and provided new insights into the pathogenic mechanism and pathological process of OGD/R. In general, this study is innovative, practical and well designed. Despite the interesting results obtained, there are some concerns and criticisms with their observations/conclusions.1. Despite the authors' comprehensive exploration of m1A modifications in neuron, they seem to have overlooked an important aspect: are m1A regulators differentially expressed in normal neurons and OGD/R neurons? The currently known m1A regulators include: Trmt6, Trmt61A, Alkbh1, Alkbh3, Fto, Ythdf1, Ythdf2 and Ythdc1. The authors could also do more in-depth literature search about the m1A regulators and performed gene expression profile analyses to identify differentially expressed regulators.

We thank the reviewer for their positive evaluation of our work and helpful comments. m1A regulators were carefully searched in published literatures. Subsequently, gene expression data of m1A regulators were extracted from the whole gene expression profile. We compared the differences in expression of those m1A regulators between normal neuron and OGD/R-treated neuron. The results showed that the expression of Alkbh3, Trmt10c, Trmt61a, Ythdf2, and Ythdf3 were statistically different (Figure 1F). However, the trends of these m1A regulators were different under different OGD/R treatments. This new data analysis now appears in Figure 1F of the revised manuscript.

2. Considering the research frontier, the biological role of m1A methylation modification should be described in detail in the Introduction section. M1A modification has been found to play a critical role in various biological conditions and diseases. Recent studies have also shown that m1A methylation regulates glycolysis of cancer cells through modulating ATP5D (PMID: 35867754, 2022) and also participate in the modulating of macrophage polarization which promotes aortic inflammation (PMID: 35620523, 2022), authors should review the latest literature on m1A modification and describe the role and significance of m1A modification in depth.

Thank you very much for the constructive suggestions and comments. We have carefully searched latest literatures about m1A modification and the revised section about the biological role of m1A methylation modification has added to the Introduction:

“Recent studies have revealed that m1A modification involved in various biological functions. Wu et al. found that m1A demethylase ALKBH3 regulated glycolysis of cancer cells and further affected tumor growth and cancer progression (Wu et al., 2022a). Wu et al. revealed that m1A regulation is significantly associated with the pathogenesis of human Abdominal Aortic Aneurysm (Wu et al., 2022b). However, the roles of m1A modification in neurons remain largely unknown.”

3. In Figure 6, the authors presented three patterns regulate m1A modification in neurons and the functions of these three patterns are identified. I consider this result to be of great interest. I suggest that the authors could further analyze the transcription factors to clarify the upstream regulation of these three patterns. Transcription factor analysis can tell us at least two things: 1. whether there is specific transcription factor expression in different patterns; 2. whether there is a regulatory network of transcription factors in different patterns.

Thank you for your suggestions. Transcription factor analysis does provide further insight into the characteristics of the three m1A patterns we identified. We extracted the genes in each pattern and predicted their transcription factor expression using the R package “decoupleR” (v 2.5.0). This new data analysis now appears in Figure 6 of the revised manuscript:

“We found that the representative transcription factors enriched specifically in cluster. Further prediction of the activity of these transcription factors showed that different transcription factor activation and repression patterns exist in different clusters. In MAC and CAC, most of the transcription factors had increased inferred activity. However, most of the transcription factors had decreased inferred activity in AAC (Figure 6F-6H). Based on these transcription factors with altered activity, we constructed a TF-target gene regulatory network (Figure 6I-6K).”

Reviewer #3 (Recommendations for the authors):Overall, this is an interesting and well performed study that described a comprehensive landscape of m1A modification in primary neuron and investigated the role of m1A in the circRNA/lncRNA‒miRNA-mRNA regulatory network following OGD/R. The focus on the two different complex regulatory networks for differential expression and differential methylation is important and it will be a valuable resource for the research community that focuses on epitranscriptomics and central nerve system diseases. Collectively, the authors present an exciting piece of work that certainly adds to the literature regarding epitranscriptomic features in neuron. While interesting results obtained and the paper is nicely written, I have several comments the authors might want to consider to improve the overall strength of their manuscript.1. In this manuscript, a meaningful conclusion is that the m1A modification on mRNA 3'UTR hinders the binding of miRNAs (Figure 5). I think this provides a good idea to study the possible roles of m1A modification in a variety of diseases. I would like to know what the authors envision for the continuation of research in this area.

Thank you for your interest in our research. To our knowledge, the m1A modifications are similar to m6A modifications and might also be cell-type-specific, so in the future we may focus on m1A modifications in other cell types of the nervous system. In addition, we will also focus on the whole process of m1A modification occurrence given that the regulatory factors of m1A are currently not studied in depth.

2. The authors present three patterns ("metabolism-associated cluster", "autophagy- associated cluster", "catabolism-associated cluster") of m1A modifications in the last part of RESULTS (Figure 6), which is an intriguing finding. However, it does not seem to explore the specific details of these three modification patterns further. I suggest that the authors could further analyze whether the genes in these patterns are regulated by some common transcription factors.

Thank you very much for the constructive suggestions. Transcription factor analysis can help us explore the characteristics of these three patterns in greater depth. We analyzed the transcription factor expression in these three patterns and added the results to Figure 6 and the corresponding Results section.

3. The m1A modification mainly depends on the dynamic regulation of RNA methyltransferases (writers), demethylases (erasers) and m1A-binding proteins (readers). However, in this manuscript the authors do not mention the changes in m1A regulators. I am curious about whether m1A regulators will be differentially expressed after OGD/R and suggest the authors to add this part of the analysis. Of course, if differentially expressed m1A regulators exist, the authors could further explore the role of these factors and, in particular, the association with the three m1A modification patterns.

We totally agree with your argumentation. This would help understand the changes of m1A modification in normal neurons and OGD/R-treated neurons. The results showed that the expression of Alkbh3, Trmt10c, Trmt61a, Ythdf2, and Ythdf3 were statistically different (Figure 1F). We have added the gene expression analysis of m1A regulators in the revised manuscript.

4. The authors obtained interesting results at the two time points set in the OGD/R model, they should try to explain the possible effects of such changes on neuronal biological processes in Discussion.

Thank you for the suggestion. In the Discussion section, we add our understanding of the m1A modifications at different time points:

“It has been shown that m6A modifications are time-specific in developmental processes as well as in the development of some diseases. During both *Drosophila* development and mouse cerebellum development (Lence et al., 2016; Ma et al., 2018), the level of m6A modification changes dynamically with the developmental process, which is associated with various organogenesis and specific functions. In Alzheimer's disease, m6A modification levels also change as the disease progresses (Shafik et al., 2021). However, whether the m1A modification is time-specific is still unknown. In the present study, we focused on the changes of m1A modification under different OGD/R treatments. Our results suggested that m1A regulators differentially expressed and mRNA and noncoding RNA has its own features in different OGD/R treatments. We suggested that this is partly indicative of the time-specificity of m1A modifications under different OGDR treatments, but more investigations are needed to determine whether m1A modifications are time-specific during development and in other disease progressions.”

5. Figure 3D and Figure 4D: The title of the Y-axis might be mislabeled.

We apologize for our error in these figures, this has now been corrected.

6. Figure 4 and Figure 6 has an additional caption at the bottom of the image, please double check.

Thank you for your careful checks. Additional captions have been removed from the Figure 4 and Figure 6.

7. P28 Line547 and P30 L575: The abbreviations of reference genome are inconsistent.

We are sorry for the consistency about the version of reference genome. It has been corrected in the revised manuscript:

“The high-quality reads were aligned to the mouse reference genome (mm10).”